# The self-inhibitory nature of metabolic networks and its alleviation through compartmentalization

Mohammad Tauqeer Alam[1,2], Viridiana Olin-Sandoval[1,3], Anna Stincone[1,†], Markus A. Keller[1,4], Aleksej Zelezniak[1,5,6], Ben F. Luisi[1] & Markus Ralser[1,5]

Metabolites can inhibit the enzymes that generate them. To explore the general nature of metabolic self-inhibition, we surveyed enzymological data accrued from a century of experimentation and generated a genome-scale enzyme-inhibition network. Enzyme inhibition is often driven by essential metabolites, affects the majority of biochemical processes, and is executed by a structured network whose topological organization is reflecting chemical similarities that exist between metabolites. Most inhibitory interactions are competitive, emerge in the close neighbourhood of the inhibited enzymes, and result from structural similarities between substrate and inhibitors. Structural constraints also explain one-third of allosteric inhibitors, a finding rationalized by crystallographic analysis of allosterically inhibited L-lactate dehydrogenase. Our findings suggest that the primary cause of metabolic enzyme inhibition is not the evolution of regulatory metabolite–enzyme interactions, but a finite structural diversity prevalent within the metabolome. In eukaryotes, compartmentalization minimizes inevitable enzyme inhibition and alleviates constraints that self-inhibition places on metabolism.

[1] Department of Biochemistry and Cambridge Systems Biology Centre, University of Cambridge, 80 Tennis Court Road, Cambridge CB2 1GA, UK. [2] Division of Biomedical Sciences, Warwick Medical School, University of Warwick, Gibbet Hill Road, Coventry CV4 7AL, UK. [3] Department of Food Science and Technology, Instituto Nacional de Ciencias Médicas y Nutrición Salvador Zubirán, Vasco de Quiroga 15, Tlalpan, 14080 Mexico City, Mexico. [4] Division of Human Genetics, Medical University of Innsbruck, Peter-Mayr-Straße 1, 6020 Innsbruck, Austria. [5] The Molecular Biology of Metabolism Laboratory, The Francis Crick Institute, 1 Midland Rd, London NW1 1AT, UK. [6] Department of Biology and Biological Engineering, Chalmers University of Technology, Kemivägen 10, 41296 Gothenburg, Sweden. † Present address: Discuva Ltd, The Merrifield Centre, Rosemary Ln, Cambridge CB1 3LQ, UK. Correspondence and requests for materials should be addressed to M.R. (email: markus.ralser@crick.ac.uk).

The activity of metabolic enzymes depends on their structure, but also on their chemical environment and non-catalytic metabolite–enzyme interactions with inhibitors and activators[1,2]. Metabolic enzyme inhibition enables feedback and feedforward loops important in the regulation of metabolism[2–5]. Some of the best studied examples are found within central metabolism[6], and include the production of a unique regulatory metabolite, fructose-2,6-bisphosphate that controls glycolysis[7,8], or fructose-1,6-bisphosphate which, as a metabolite correlated with flux[3], acts as an allosteric feed-forward activator of glycolysis ATP net-producing enzyme, pyruvate kinase (PK)[9].

However, the evolutionary origins of metabolic enzyme inhibition are neither, in essence, explained by the need to regulate metabolism, nor are the general principles that guide metabolic enzyme inhibition. Metabolic feedback inhibition for instance can be a direct consequence of the catalytic mechanisms itself[10]. In other instances, distally produced metabolites act as inhibitors by having strong structural similarity with the enzymatic substrates. For instance, phosphoenolpyruvate inhibits triosephosphate isomerase (TPI) due to extensive structural similarity with dihydroxyacetone phosphate, which constrains the activity of glycolysis when cells respire[5,11]. This and other self-regulatory metabolite–enzyme interactions have been incorporated into mathematical models, upon which a much better quantitative reflection of metabolic functionality is achieved[12–15]. Enzymes hence participate in a fully functional metabolism while being partially inhibited.

As studying enzyme inhibition remains a laborious and mostly *in vitro* process, there is little information about its global nature. In order to obtain insights into the principles of metabolic self-inhibition, we curated and combined comprehensive enzymological knowledge obtained over a century of biochemical research, that has been collected in the Braunschweig Enzyme Database (BRENDA) database[16] with a genome-scale reconstruction of the human metabolism[17]. We obtained a highly structured enzyme-inhibition network that shows inhibition is predominantly emerging from chemical structural constraints. With different metabolite chemical specimen affecting specific enzyme classes, metabolic enzyme inhibition is found to be an extremely frequent phenomenon that affects virtually all biochemical process. We provide evidence that metabolic enzyme inhibition constraints metabolism to the extent that cells evolve to minimize unwanted metabolite–enzyme interactions. A key mechanism is identified with the specific organellar localization of enzymes in eukaryotic cells, that prevents an enrichment of metabolic–enzyme inhibition within the organellar metabolic neighbourhood.

## Results

**A metabolome-scale enzyme-inhibition network.** Over a period of nearly 30 years, the BRENDA database[16] has collected 201,940 reference citations and 170,794 enzyme entries associated with 6,763 Enzyme Commission (EC) classes (as of July 2015), summarizing enzymology data that dates back to the beginning of biochemical research. After curation of this data set, we were able to map 30,107 inhibitors to 685 of 747 (91.7%) biochemical reactions (EC numbers) contained in the human metabolic network reconstruction Recon2 (ref. 17). The median number of inhibitors per reaction is 29, spanning over a broad range. Sixteen enzymes have just one inhibitor assigned, while the most extreme case is monoamine oxidase (EC:1.4.3.4) with 1,017 inhibitors.

Next, we removed all entries that could not be matched to a unique metabolite (KEGG, HMDB identifiers), and restricted the network to those inhibitors that are human metabolites.

This core of the data set consists of 1,311 studies which were specifically conducted on 333 human enzymes and 431 human metabolites. Then, we expanded the network to include 'cross-species' data, exploiting the fact that the principles which guide enzyme inhibition are the same across species (Fig. 1a). In this way we obtained a network in sufficient coverage to derive general principles. Indeed, inhibitors are in most cases not specific to one species (that is, refs 18–21), and we revealed that more than half (693/1,311 (53%)) of human metabolic inhibitors were experimentally reported in at least one other species as well. In the final network, 10.3% of all edges are based on human-only experiments, and 21.9% are derived from inhibitors reported on both the human enzyme as well as at least one other species.

Retaining the topological organization and chemical make-up of the human metabolic network, the cross-species informed inhibitor network consists of 682 metabolic inhibitors (mapping to 26% of human metabolites) that inhibit 83% (621) of its distinct enzymatic reactions (Fig. 1a,b, Supplementary Data 1). The bipartite inhibition network consists of 5,989 edges associated with 1,303 nodes. It possesses scale-free structure (degree distribution empirically follows a power-law distribution with parameter $2 < \gamma < 3$, compared to a random network where $\gamma = 11.47$; Fig. 1c) and has characteristics of a typical non-random biological network. The most connected inhibitor is ATP (maximum connectivity 167), while 90% of enzymes and inhibitors have 20 or less connections (Fig. 1c).

Owing to the rich amount of data, each enzyme EC category is similarly well covered by inhibitor information. In all, 208 out of 244 transferases are inhibited by at least one metabolite, 159 out of 205 oxidoreductases, 147 out of 161 hydrolases, 49 out of 66 lyases, 32 out of 39 isomerases, and 26 out of the 28 ligases (Fig. 1c,d, Supplementary Table 1). All enzyme classes are found similarly susceptible to metabolic inhibition; most inhibited class of enzymes was also the largest enzyme class, transferases (1964, or 32.8%, of all inhibitor interactions) followed by oxidoreductases (1656; 27.7%), hydrolases (1388; 23.2%), lyases (496; 8.3%), ligases (293; 4.9%), and isomerases (188; 3.1%) (Supplementary Table 1).

To classify the inhibitors, we adopted the chemical categorizations of the human metabolome database (HMDB 3.5)[22], merging small superclasses to yield eight groups of chemical similarity (Supplementary Table 2): 'Lipids' (178 out of the 610 lipids) account for most inhibitions, followed by 'Aromatic Cyclic Compounds' (105 out of 161), 'Amino Acids, Peptides and Analogues' (85 out of 176), 'Nucleosides, Nucleotides and Analogues' (84 out of 102), 'Carbohydrates and Carbohydrate Conjugates' (63 out of 98), 'Organic Acids and Derivatives' (43 out of 69), 'Aliphatic Acyclic Compounds' (40 out of 65) plus all 'other compounds' (84 out of 376) (Fig. 1d, Supplementary Table 2). Twenty-nine per cent (197/682) of the inhibitors were phosphorylated. This includes 83.3% of 'Nucleosides, Nucleotides and Analogues', 36.5% of 'Carbohydrates and Carbohydrate Conjugates', 26.4% of 'Lipids' and 3.5% of 'Amino Acids, Peptides and Analogues' category (Supplementary Table 2).

As metabolites can inhibit multiple enzymes, the number of inhibitory interactions is substantially different to the number of inhibitors per metabolite class. Thirty per cent of all inhibitor interactions are accountable to 'Nucleosides, Nucleo-tides, and Analogues' (1,749 inhibitory interactions), followed by 'Lipids' (810; 13.5%), 'Amino Acids, Peptides, and Analogues' (613; 10.2%), 'Aromatic Cyclic Compounds' (592; 9.9%), 'Aliphatic Acyclic Compounds' (541; 9%), 'Organic Acids and Derivatives' (537; 9%) and 'Carbohydrates and Carbohydrate Conjugates' (375; 6.2%), and a total of 721 (12.9%) by other metabolites (Supplementary Table 2). The dominance of

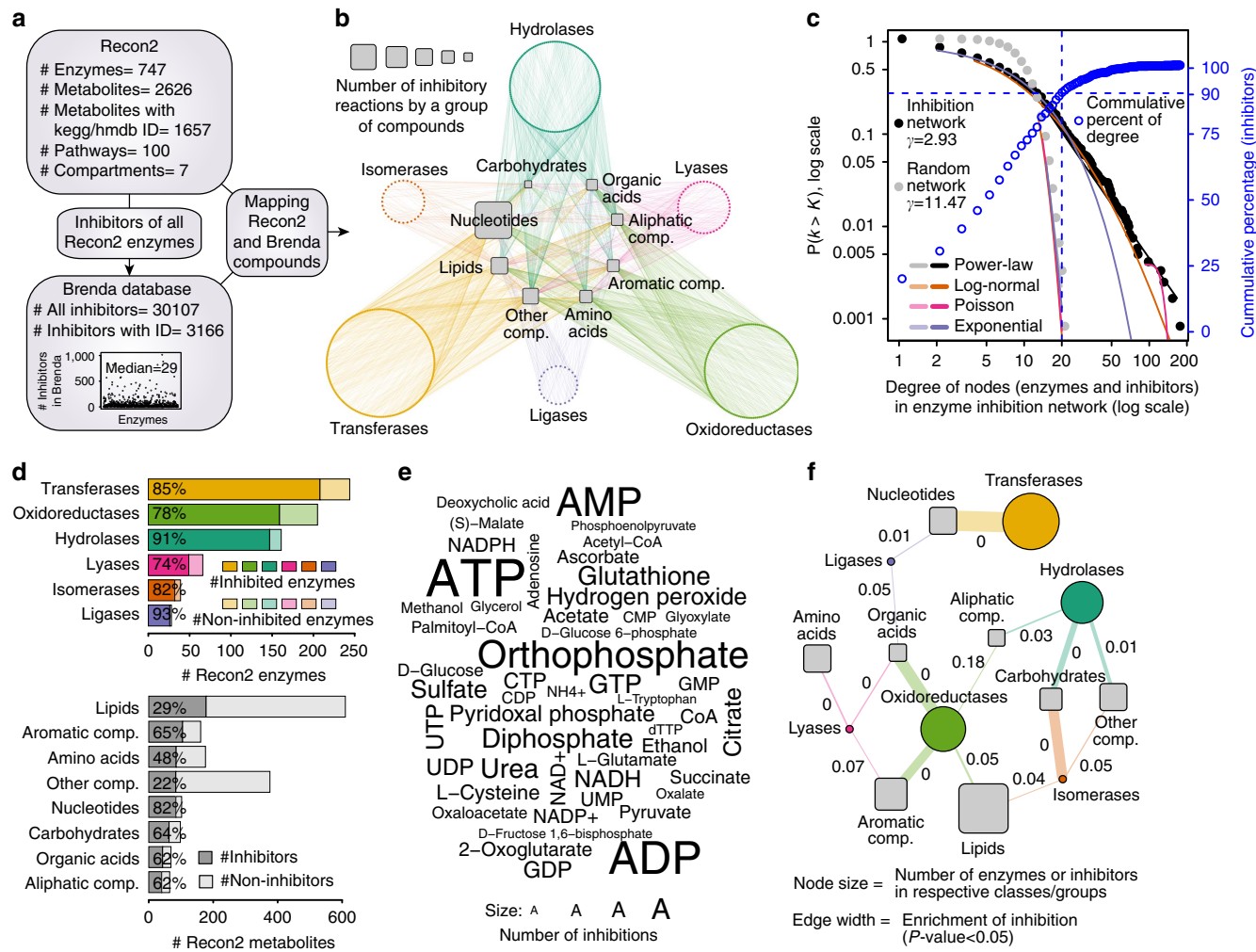

**Figure 1 | A genome-scale network of enzyme inhibition.** (**a**) construction of a genomic-scale enzyme-inhibition network by mapping inhibitor information curated from the BRENDA database[16], to the human metabolic reconstruction (Recon2 (ref. 17)). (**b**) enzyme-inhibition network (non-directional illustration), in which 82% of enzymatic reactions are inhibited by 26% of Recon2 metabolites. Enzymes are coloured and grouped according to enzyme commission (EC) category, inhibitors according to their HMDB chemical classification[22], node size is scaled numerically. (**c**) The enzyme-inhibition network is scale-free, and follows a power-law and a log-normal distribution (P value for comparing power-law distribution with log-normal, Poisson and exponential are 0.36, 8.7e-05, 0.077 respectively), in comparison to a random network of the same size which is not scale-free (P value for comparing power-law distribution with log-normal, Poisson and exponential are 0.42, 0.78, 0.44, respectively). Ninety per cent of enzymes and metabolites have 20 or less connections (blue line). (**d**) Top: Enzyme classes (EC classifications) according to their occurrence in the genome, in relation to their representation in the inhibition network. Bottom: Metabolites categorized according to HMDB superclass[22], and the percentage, and to which extent they are inhibitors in **b**. (**e**) Fifty most frequently inhibiting metabolites illustrated as word-cloud, scaled to the number of inhibitory interactions annotated for each inhibitor. (**f**) Enzyme classes are inhibited dependent on the metabolite's chemistry. Size of the nodes is scaled according the number of inhibitor/enzymes within each class. The edge thickness is scaled according to the occurrence of significant inhibitory interactions between the inhibitor's superclass, and enzyme class. Nodes are connected if P value < 0.05. FDR values are highlighted over edges. Abbreviations refer to HMDB categories: *Amino acids*, Amino Acids, Peptides and Analogues; *Aliphatic comp.,* Aliphatic Acyclic Compounds; *Aromatic comp.,* Aromatic Cyclic Compounds; *Carbohydrates*, Carbohydrates and Carbohydrate Conjugates; *Organic acids*, Organic Acids and Derivatives; *Nucleotides*, Nucleosides, Nucleotides and Analogues.

'Nucleosides, Nucleotides, and Analogues' is also reflected on the level of the most potent single inhibitors, adenylate and nicotinamide nucleotides (Fig. 1e, Supplementary Data 1). While these absolute numbers are subject to testing bias, a statistical analysis that corrects for the sample size confirmed the dominant role of nucleotide inhibitors, that are not only the most frequent inhibitors but also the most metabolically connected metabolites. Importantly, the chemical identity of the inhibitors was found to reveal the enzyme class they most likely inhibit (Fig. 1f, Supplementary Tables 3–5). 'Nucleosides, Nucleotides, and Analogues' are significantly enriched to inhibit Transferases and Ligases (P value 4.8e − 34 and 0.01 respectively), 'Amino Acids,

Peptides, and Analogues' to inhibit Lyases (P value 1.8e-03), 'Carbohydrates and Carbohydrate Conjugates' to inhibit Isomerases and Hydrolases (P values 5.9e-18 and 6.7e-11, respectively); 'Lipids' inhibit Oxidoreductases and Isomerases (P values 2.3e-04 and 2.9e-02); 'Organic Acids and Derivatives' inhibit Oxidoreductases, Ligases and Lyases (P values 5.3e-20, 3.1e-02 and 6e-03), 'Aliphatic Acyclic Compounds' inhibit Oxidoreductases and Hydrolases (P values 2e-2 and 1.2e-3); and finally 'Aromatic Cyclic Compounds' inhibit Oxidoreductases and Lyases (P values 1.5e-15 and 2.9e-2) (Fig. 1f, Supplementary Tables 3–5). Despite the inhibition network includes also weak inhibitors as stored in BRENDA, and its underlying

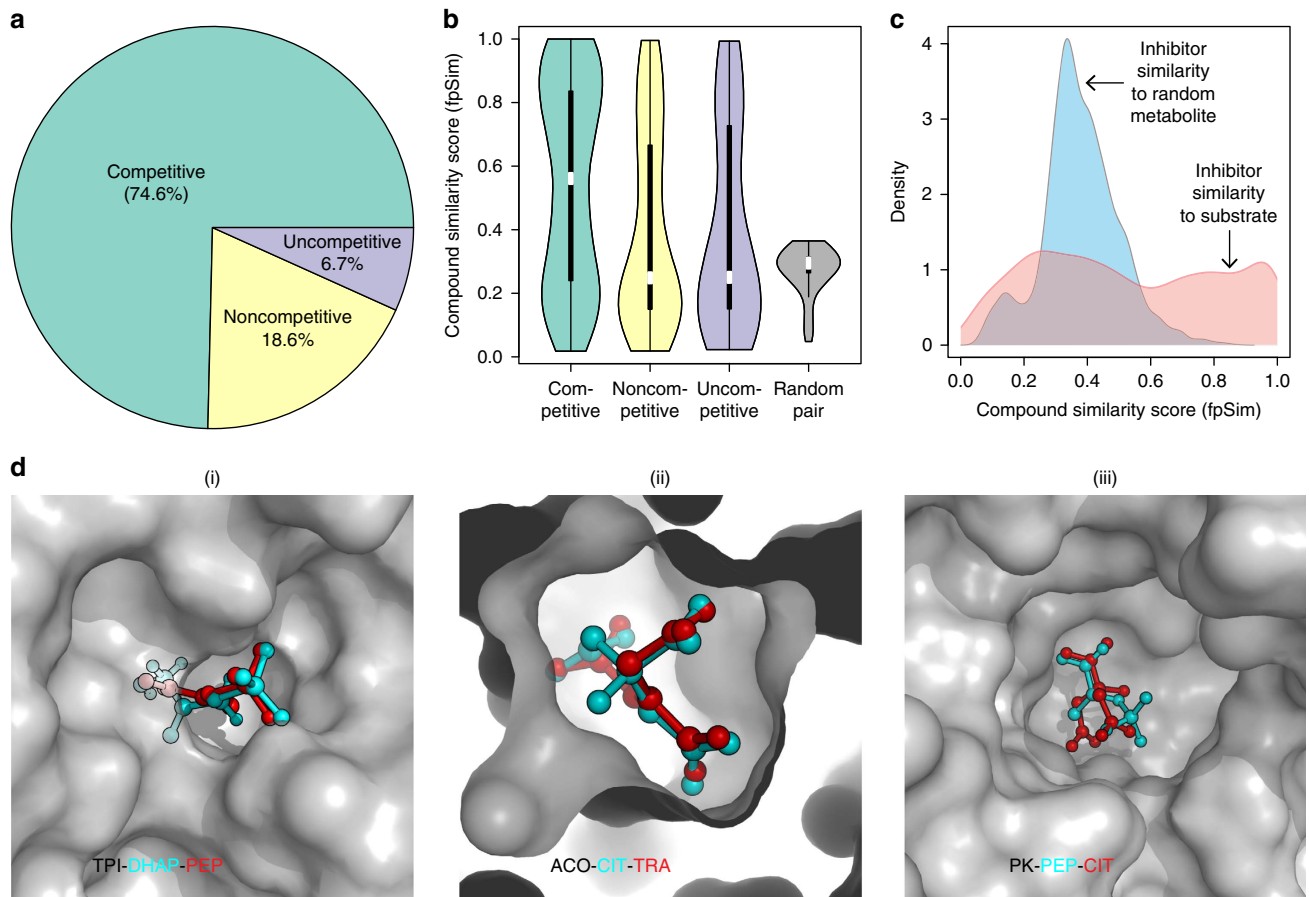

**Figure 2 | Enzyme inhibition across the metabolic landscape is driven by structural similarity between metabolites.** (**a**) Gold-standard set: Competitive inhibition is the most common type of inhibition (74.6%) followed by noncompetitive (18.6%) and then by uncompetitive inhibition (6.7%), across 462 examples determined in individual enzymological experiments (Supplementary Data 1). (**b**) Pairwise compound similarity (fpSim) between inhibitor and substrate reveals significant structural similarity (0 = non-similar, 1 = maximum possible structural similarity). The median similarity for allosteric inhibitors is not significantly different to that of a random inhibitor; however, the spread is much higher, with a about 1/3rd of allosteric inhibitors being as similar enzyme's metabolic substrates as competitive inhibitors. (**c**) The genome-scale set of enzymatic inhibitors have a wide range of similarities including highly similar metabolite–inhibitor relationships (ranging from 0.2–1), compared to random substrates, which are not structurally similar (0.2–0.4). (**d**) Three selected examples (protein/metabolite structure determined by X-ray crystallography) revealing the extent of structural similarity between competitive inhibitor and substrate in the active site of key enzymes. (i) Triosephosphate isomerase (TPI: 1NEY, inhibitor: 4OWG (ref. 5)); Substrate: Dihydroxyacetone phosphate (DHAP); Inhibitor: phosphoenolpyruvate (PEP), (ii) Enzyme: Aconitase (ACO: 1C96 (ref. 28), inhibitor: 1ACO (ref. 29)); Substrate: Citrate (CIT); Inhibitor: Trans-aconitate (TRA), (iii) Enzyme: Pyruvate Kinase (PK: 4HYV (ref. 26), inhibitor: 4IP7 (ref. 27)); Substrate: phosphoenolpyruvate (PEP); Inhibitor: Citrate (CIT).

data are subject to literature bias, the network reveals highly distinct topology that is dependent on the chemical class of the metabolites.

**Competitive inhibition dominates metabolism.** The biochemical literature distinguishes three major inhibitory mechanisms: competitive, noncompetitive and uncompetitive (or metabolic allostery). Of these, competitive inhibition implies indeed a structural relationship between inhibitor and substrate[23,24]. We questioned which mechanisms would dominate. We manually curated the data set to create a 'gold standard' of single-experiment determined reaction mechanisms. Competitive inhibition accounted for 74.6% of the gold-standard mechanisms, noncompetitive for 18.6% and uncompetitive inhibition for merely 6.7% (Fig. 2a). We selected three representative examples TPI, aconitase and PK, for which crystallographic structures both for the enzymes bound to substrate and competitive inhibitor are available[5,25–29]. In all three cases, the

structures revealed striking structural similarity with a metabolic substrate (Fig. 2d).

Next, we used computational structure prediction to enumerate similarities between substrates and inhibitors. Pairwise compound comparisons with fingerprint similarity (fpSim[30]), drug molecule similarity derived by molecular fingerprints (calcDrugFPSim[31]), maximum common substructure (Tanimoto Coefficient, Overlap Coefficient[32]) as well as pairwise compound comparisons with atom pairs (cmp.similarity[30]), all revealed that the typical competitive inhibitor possesses significant similarity to at least one enzymatic substrate in the gold-standard data set (Fig. 2b *P* value 2.84e-06, Supplementary Fig. 1).

For most inhibitory interactions that constitute our network the mechanism is unknown. However significant similarity between inhibitor and at least one substrate is detected across the entire network. Vice versa, even the most similar metabolite out of a poolsize-corrected random sampling lacks structural similarity (Fig. 2c, Supplementary Fig. 2). Most of metabolic enzyme inhibition is hence explained by

**Table 1 | Crystallographic data and refinement statistics.**

| Ligand | NADH + oxaloacetate | Malonate |
|---|---|---|
| Space group | P2$_1$ | P2$_1$ |
| Cell dimensions | | |
| a, b, c (Å) | 72.373, 138.763, 74.567 | 70.806, 76.952,118.858 |
| α, β, γ (°) | 90, 110.12, 90 | 90, 96.48, 90.0 |
| Resolution (Å) | 1.87 | 1.58 |
| $R_{merge}$, % | 10.3 (75.1) | 6.9 (100.2) |
| Total number of observations | 741,462 (42,498) | 318,978 (18,851) |
| Unique reflections | 112,079 (7,366) | 155,035 (9,932) |
| Completeness, (outer shell)% | 98.9 (87.7) | 89.7 (78.5) |
| I/σI (outer shell) | 14.00 (2.70) | 8.00 (0.80) |
| <I> half-set correlation | 99.7 (56.9) | 99.7 (33.4) |
| Reflections used for refinement | 112,037 | 147,320 |
| Geometry: Ramachandran favoured, allowed, outliers, % | 98.18, 1.82, 0.0 | 98.18, 1.82, 0.0 |
| Rms deviation from ideal values for bond lengths (Å) and bond angles (°) | 0.012/1.227 | 0.018/1.734 |
| Number of non-hydrogen atoms in refinement | 11,166 | 11,632 |
| Refinement ($R/R_{free}$ 5% reflections in test set), % | 17.00/19.90 | 17.40/20.30 |

structural similarity between substrates and other cellular metabolites.

Enzymes have not been exhaustively tested against all metabolites. We hence computed structural similarity coefficients for all metabolite pairs within the metabolic network. Considering the average structural similarity between substrate and inhibitor as cut-off, 97.7% of human metabolites (1,338 out of 1,369 compounds, with Kegg/HMDB ID) are structurally similar to at least one other human metabolite. A confined chemical structural diversity within the metabolome makes in essence every enzymatic reaction susceptible to competitive inhibition.

**Many allosteric inhibitors resemble the substrate structure.** Non-competitive and uncompetitive enzyme inhibition is not dependent on structural similarity between substrate and inhibitor, as the inhibitor does bind to a different site in the enzyme. Nonetheless, 33% of non-competitive and 31% uncompetitive inhibitors are structurally similar to the enzyme's substrates (Fig. 2b, Supplementary Fig. 1). One illustrative example was identified with L-lactate dehydrogenase (L-LDH). L-LDH is allosterically inhibited by its substrate, pyruvate[33,34]. However, also malonate, not a typical intermediate of mammalian metabolism, is a non-competitive inhibitor[35]. Further, oxaloacetate has been reported to inhibit L-LDH as well[36–38]. We performed computational similarity calculations, and found that pyruvate, malonate and oxaloacetate all are structurally highly similar (Malonate − pyruvate similarity = 0.667, substructure overlap = 0.834; Oxaloacetate − pyruvate similarity = 0.685, substructure overlap = 1; and Malonate − Oxaloacetate similarity = 0.685, substructure overlap = 0.857 (similarity = 'Pairwise compound comparisons with PubChem fingerprints'[30]; substructure overlap = 'Maximum common substructure Overlap Coefficient')[32].

In order to test whether structural similarity could explain the allosteric inhibition, L-LDH purified from rabbit muscle was co-crystallized with oxaloacetate and NADH, or malonate, respectively. The structures were solved by X-ray crystallography followed by molecular replacement, and refined at 1.87 and 1.58 Å resolution, respectively (refinement statistics in Table 1). Malonate and oxaloacetate bind L-LDH in a similar fashion as the structurally similar pyruvate. L-LDH forms a tetramer and a complete tetramer occupies the asymmetric unit for both crystals (Fig. 3a). Analysing the electron density map of L-LDH co-crystallized with oxaloacetate and NADH, we could identify oxaloacetate at partial occupancy in two of the four subunits (Fig. 3a). Oxaloacetate forms a hydrogen bond network with R105, R168, H192, N137, T247 and numerous water molecules. Electron density for malonate at partial occupancy was present in three of the subunits for the crystal grown without the cofactor NADH (Fig. 3a). Both inhibitors bind adjacent to, but not within, the active site (Fig. 3b,c) and interact with the same amino acids (Fig. 3d). Moreover, electron densities corresponding to two malonates were present in the interface between chains A and C and between B and D. These ligands form hydrogen bond interactions with R170 from one chain and S183, H185 of the other chain and various water molecules. However, no inhibitor was found in the interface of L-LDH when co-crystallized with oxaloacetate and NADH. Instead, sulfate ions were present in the same location interacting with R170, H185 of one chain and H185 of the other (Fig. 3a).

Next, we determined the kinetic mechanism *in vitro* and found L-LDH inhibition by malonate to match the model of non-competitive inhibition ($R^2$ of 0.93). The $Km$ obtained for pyruvate and NADH were 0.300 ± 0.035 and 0.095 ± 0.012 mM, respectively. Moreover, we find that, similar to malonate, oxaloacetate is a non-competitive inhibitor of L-LDH with respect to pyruvate with moderate *in vitro* inhibition too ($Ki = 2.30 \pm 0.22$ mM) (Fig. 3e). Even though oxaloacetate is not a strong inhibitor, it has a higher inhibitory capacity ($Ki_{OAA}/Km_{Pyr} = 7.7$) compared to malonate ($Ki_{Malon}/Km_{Pyr} = 90$; considering a $Ki = 27$ mM (ref. 35)). Although it is unlikely that malonate (and perhaps oxaloacetate) play an important role in the regulation of rabbit L-LDH activity, it allosterically inhibits the enzyme, and this interaction is explained by its structural similarity with pyruvate.

**Metabolic network neighbours are most likely inhibitors.** Metabolites which are metabolically derived from each other are structurally more likely similar as compared to more distal metabolites that underwent more enzymatic conversions (Fig. 4a). Indeed, structural similarity clusters strongly reflected topological distances within the inhibition network (Fig. 4b, P value 1.64e-99, t-test). Structural analogues are hence typically found within the close metabolic neighbourhood (Fig. 4b, Supplementary Fig. 3). Consistently, we find that the closer the metabolite is topologically to an enzyme, the more likely it is an inhibitor (Fig. 4c, P value 1.91e-14, t-test). As a consequence,

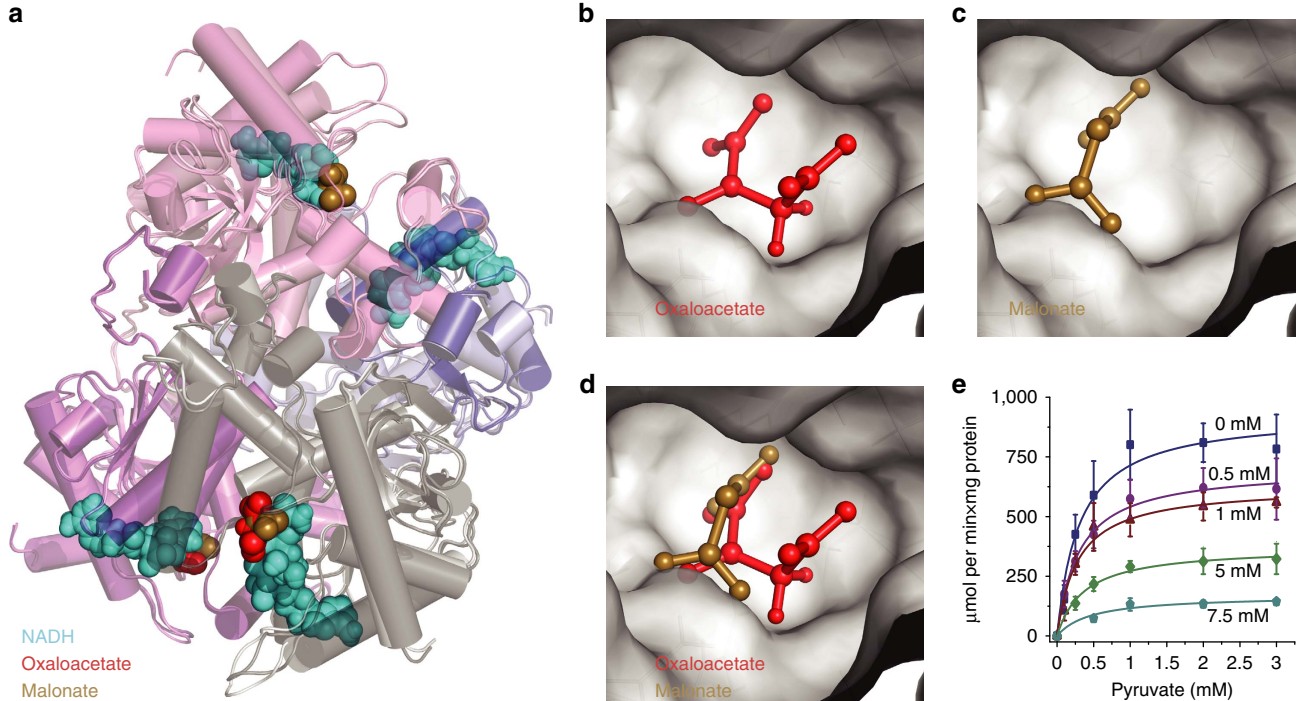

**Figure 3 | Structural similarity to pyruvate renders malonate and oxaloacetate allosteric inhibitors of L-lactate dehydrogenase.** (**a**) Alignment of the X-ray crystallographic structures generated for rabbit muscle L-LDH co-crystallized with oxaloacetate (red) and NADH (cyan); or with malonate (gold). Active site of L-LDH with (**b**) oxaloacetate (OAA), or with (**c**) malonate. (**d**) These inhibitors are structural analogues of pyruvate and are found to bind all to the same non-competitive site. (**e**) Oxaloacetate is a non-competitive inhibitor of L-LDH with respect to pyruvate. Enzyme kinetics were determined spectrophotometrically ($n = 3$) and fit according to a non-competitive model ($R^2 = 0.93$). Error bars = mean ± s.d.

inhibitors are enriched to occur within metabolic pathways (Supplementary Figs 4,5). Metabolic feedback and feed-forward loops emerge hence most likely within metabolic pathways for structural reasons.

**Most connected metabolites are frequent inhibitors.** Glycolysis, gluconeogenesis and nucleotide interconversions were the most inhibited and inhibiting metabolic pathways (Fig. 4d,e). While this particular result could be the consequence of a testing bias, we confirmed a highly significant correlation between the number of inhibitory interactions and metabolic reactions for metabolites across the network, revealing that central metabolites are the most frequent inhibitors (Fig. 4f, $P$ value 2.2e-16, $r$ 0.627, Pearson's correlation). Indeed, the more central or essential a metabolic enzyme, the more likely its substrates are inhibitors (Fig. 4g, $P$ value = 1.71e-11, $t$-test). Furthermore, all of the most frequent inhibitors were essential metabolites (Fig. 4h), and metabolites that function across subcellular compartments inhibit far more enzymatic reactions, as other metabolites (Fig. 4k). Indeed, this finding is metabolically plausible for the reason that these metabolites are also the frequent substrates. We asked whether metabolite essentiality and structural similarity could be one explanation. Indeed, essential metabolites were more structurally similar to each other, compared to non-essential metabolites, which reflects their central localization in the network (Supplementary Fig. 6). Such a situation could emerge if evolution selects against enzyme inhibitors; a negative selection would be much more effective on non-essential metabolites compared to essential metabolites. Indeed, essentiality was found to be only a factor for metabolites to be inhibitors but not for enzymes to be inhibited. Although enzymes are more likely to be essential when they are implicated in more metabolic reactions[39],

they were inhibited irrespectively of whether they are essential or are not essential (Fig. 4i,j).

**Subcellular compartmentalization reduces enzyme inhibition.** Metabolic organization in the seven human (organellar) compartments may reduce unwanted enzyme–inhibitor interactions[40]. We started with simulating random compartmentalization. Allowing enzymes and inhibitors to participate in multiple compartments as in the human metabolic network, the number of inhibitory interactions is reduced as more hypothetical compartments are introduced, an effect that saturates after 8–10 compartments (Fig. 5a). In a network that omits multiple localized metabolites, the effects are more dramatic (Supplementary Fig. 7a).

The metabolic network can however not randomly compartmentalize due to its chemical connectivity. We therefore examined the effect of compartmentalization upon including enzyme localization to cytoplasm, Golgi apparatus, lysosome, mitochondria, nucleus, endoplasmic reticulum (ER) and peroxisome to the inhibitor network. The maximum number of inhibitory interactions takes place within cytoplasm followed by mitochondria, the two compartments that host also the highest number of metabolic reactions (Fig. 5b). Accounting for multiple metabolite localizations, the total number of inhibitory interactions is found significantly reduced for organelle-localized reactions, even after correcting for their smaller size (Fig. 5c,d). Only 6% of cytoplasmic inhibitors loose co-localization with the enzyme they inhibit, but 38% of mitochondrial, 54% of ER, 65% of peroxisomal, 59% of the nuclear, 67% of lysosomal and 71% inhibitors produced in the Golgi (Fig. 5d). As a consequence, compartmentalization reduces 2% of inhibitory interactions within the cytoplasm, but 21% in mitochondria, 43%, in the ER, 49% in the peroxisome, 46% in the nucleus, 60% in the lysosome and 54% in the Golgi. A specific

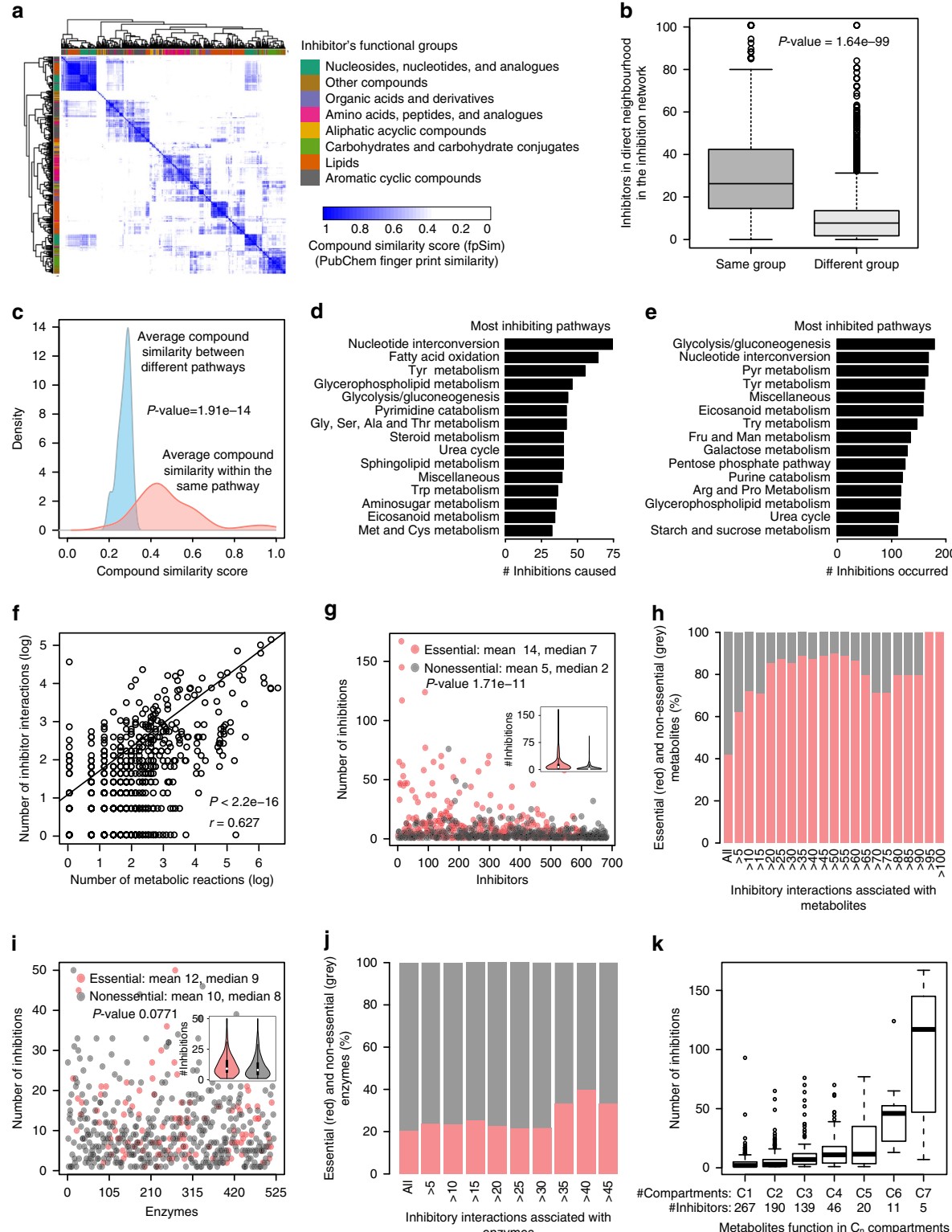

**Figure 4 | Inhibitors emerge in the metabolic neighbourhood of enzymes and are often the essential and most central metabolites.** (**a**) Human metabolites grouped according to the HMDB superclass upon pairwise compound structural comparisons by fingerprint similarity (fpSim). (**b**) Inhibitors neighbour other inhibitors of the same HMDB group in the inhibition network. (**c**) Metabolites of the same metabolic pathway as the enzyme's substrates possess significant structural similarity, rendering them the most likely inhibitors. (**d**) Fifteen top most inhibiting pathways in the inhibitor network. (**e**) Fifteen top most inhibited pathways (full list in Supplementary Data 1). (**f**) The number of inhibitory interactions and number of metabolic reactions per metabolite correlate across the metabolic network. (**g**) Metabolites participating in an essential biochemical reaction significantly inhibit more enzymes. (**h**) Vice versa, essentiality increases with degree of inhibition, and all of the most frequent inhibitors are essential metabolites. (**i**) The same is not the case for enzymes; essential and nonessential enzymes are similarly frequently inhibited; (**j**), and enzyme essentiality remained constant with the increase in degree of inhibition. (**k**) Metabolites that function in multiple organelles (that is: $C_n$: metabolite participates in enzymatic reactions in $n$ compartments) inhibit more reactions. $P$ values were calculated using Welch two sample $t$-test (**b,c,g,i**) or Pearson's product–moment correlation (**f**).

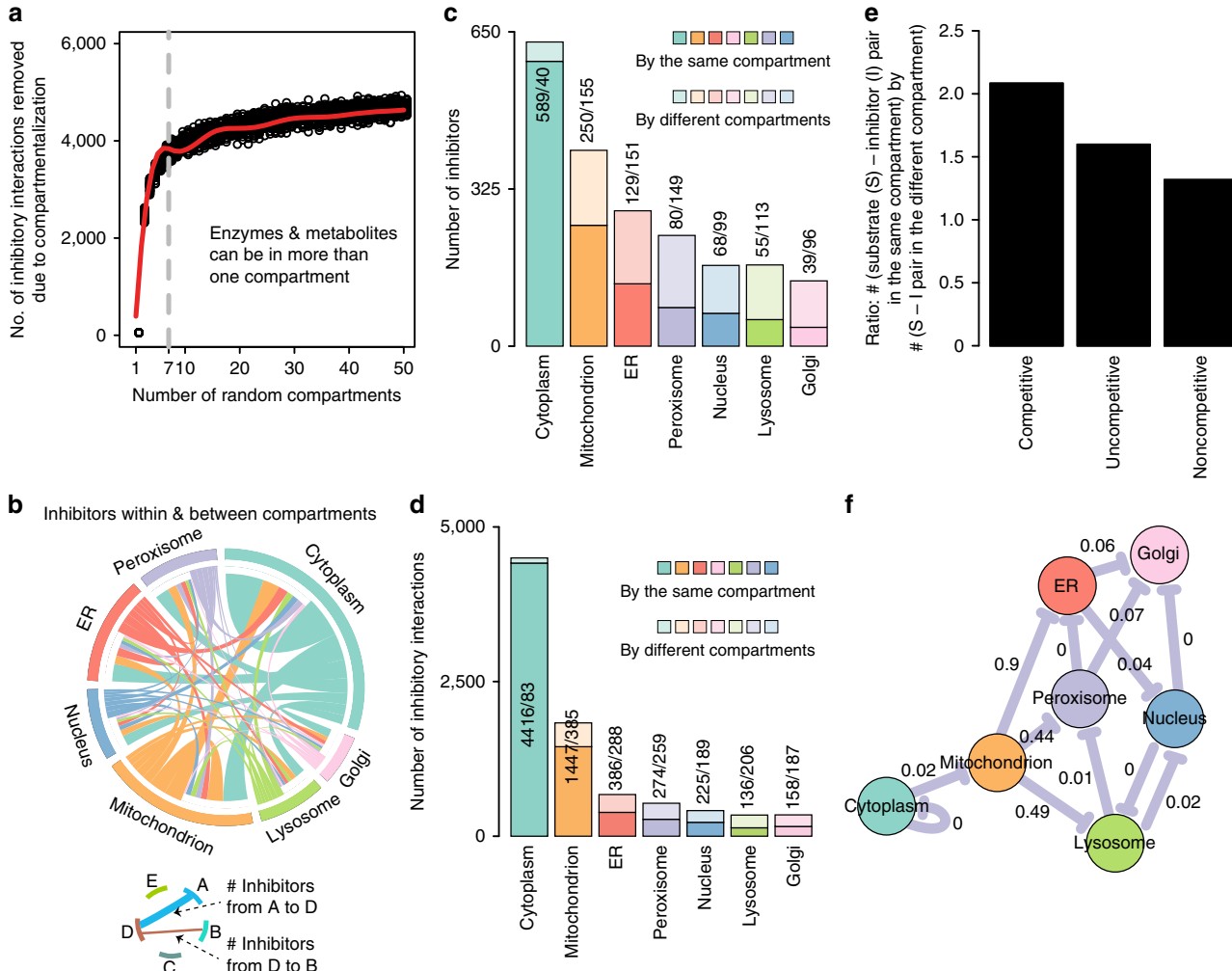

**Figure 5 | Compartmentalization considerably reduces metabolic self-inhibition.** (**a**) Computational simulation by randomly placing inhibitors and enzymes in random compartments, maintaining a metabolic network organization. red line: *bs* spline fitting; grey line: seven organellar compartments as in the humans. (**b**) Human network: compartmentalization globally reduces inhibitor–enzyme interactions. (**c,d**) Compartmentalization protects organellar metabolic reactions from enzyme inhibition (in c expressed as the number of inhibitors, in d as the number of inhibitory interactions). (**e**) Compartmentalization is more effective in preventing allosteric inhibition, as on preventing competitive inhibition. (**f**) Compartment–compartment specific inhibition network in which nodes represent the seven subcellular compartments. Edges are drawn if the number of inhibitor compounds between a pair of compartments is significantly enriched (*P* value < 0.05). FDR values are highlighted over edges. Metabolic self-inhibition is enriched in the cytoplasm as expected (Figs 1–3), but not within the organelles. Instead, organellar metabolites would significantly enrich inhibition if localized to other organelles (*P* value < 0.05).

organellar localization does explain this result. The number of inter-compartment interactions per inhibitor was constant across all compartments including cytoplasm (Supplementary Fig. 7b). However, the ratio of inter-compartment inhibition versus total inhibition is substantially lower for organellar enzymes (Supplementary Fig. 7b). If excluding mulitpe-localized metabolites, this picture further substantiates (Supplementary Fig. 7c). Next, we compared the number of inhibitors within and between compartments, and upon correcting for sample size, expressed the results as a compartment inhibition network (Fig. 5f). Only in the cytoplasm we detected the enrichment of inhibitors as expected from the structural similarity that prevails in the metabolic neighbourhoods (Fig. 4c). Inhibitors did however not enrich in any of the organelles. As a control experiment, we compared the inhibitory potential if organellar enzymatic reactions would localize to another compartment. A change of localization would result in a significant enrichment of inhibition (illustrated as inhibition between compartments) (Fig. 5f). Hence, the effect that enzyme

inhibitors are enriched within the close metabolic neighbourhood is counteracted by compartmentalization.

## Discussion

Enzyme inhibition has been studied for decades in the context of enzyme function and in the regulation of metabolism, but there is still little information about its global nature. However, enzymologist collected *in vitro* data about enzyme function in thousands of carefully conducted small-scale experiments over a century. A data set of comparable depth not available for any other biological problem. We took the assumption that the basic principles of enzyme inhibition will be the same across species. The cross-species inhibitor information that could be mapped onto the topology of the human metabolic network reconstruction—itself based on experimental results derived from multiple species—is estimated to be the result of more than 9,000 years of single-person experimental work, and our network incorporates work of > 18,000 research publications.

The large size was required to outweigh, or at least statistically define, the testing bias that exists in the biochemical literature.

The high coverage indeed alleviated a selective bias against a certain enzyme class or reaction mechanism, so that all human EC classes are equally well covered (Fig. 1). The inhibitor space instead is restricted to commercially available metabolites and hence, is less complete. Nonetheless, also here a critical number of inhibitors represent each HMDB metabolite class (Fig. 1d) and no coverage bias caused by the purchase cost of the chemicals (a proxy of its market properties) was detected (Supplementary Fig. 8). Another potential bias could be caused by the situation that competitive inhibitors are easier to be predicted compared to allosteric inhibitors; however, the number of incorporated studies that base on structural prediction is marginally small. Finally, there is a coverage bias, because popular enzymes or metabolites like ATP have been studied more often than other metabolites. Here, we apply a number of statistical methods that correct for sample size, and discuss this bias in all cases where it is relevant.

Our network is only qualitative, as we were unable to incorporate quantitative kinetic information (that is, $Ki$, $Km$ values). First, for only a fraction of the inhibitors this information is available, second our network is cross-species and enzyme constants vary between species, and third, many enzymatic constants differ substantially if determined by more than one lab (Supplementary Fig. 9). The latter situation is most likely explained by differences in experimental conditions, pH in particular. This result shows however that enzyme inhibition might form a dynamic network also biologically, as conditions like pH change also in vivo and differ between organelles. Consequently, enzyme inhibitors, even those that have strong inhibitory constants, might be biologically relevant only under a subset of conditions, while supposedly weak inhibitors could conditionally become relevant inhibitors. In any case, weak inhibitors have to be considered a metabolic constraint due to the fact that thousands of them act in parallel.

Metabolite–enzyme inhibition is required to regulate enzyme function. Our data indicate however that it is not necessarily evolved for this purpose, and that only a fraction of metabolite–enzyme interactions are indeed of regulatory nature. Instead, a majority of inhibitory interactions seem to emerge as a property caused by metabolite stereochemistry. Most of enzyme inhibition is to be traced to metabolites that possess significant structural similarity to the enzymes metabolic substrates, and we find that inhibitors enrich as a function of the metabolic network's topological distance. Notably, our data show that not only competitive inhibition, but also allostery is affected by chemical structural constraints. One-third of allosteric inhibitors do possess significant structural similarity to the inhibited enzyme's substrates. Identifying an illustrative case with L-LDH, we determined its molecular structure by X-ray diffraction when bound to two non-competitive metabolites. We found that both inhibitors, one of them not common to the mammalian cell, bind to L-LDH as a consequence of their structural similarity to a biological relevant L-LDH regulator, pyruvate.

A recent and illustrative example for metabolic constraints caused by enyzme inhibition concerns glyceraldehyde-3-phosphate dehydrogenase and PK, that promiscuously produce 4-phospho-erythronate and 2-phospho-L-lactate. These metabolites need to be removed by specific clearance enzymes in order that glycolysis and pentose phosphate pathway (PPP) are not inhibited[41]. Enzyme inhibition can have hence negative consequences for metabolism to the extent inhibitor clearance is frequently observed as a part of 'metabolite repair' processes that are energetically costly[42]. One may argue however that many enzyme inhibitors have relatively weak inhibitory constants.

One has to take into consideration however that the metabolites which are part of the human metabolic network are the ones that prevailed, while strong inhibitors as the ones aforementioned were negatively selected and have not been retained. Consistently, across the network, we find that a highest number of inhibitions is observed for the most important and most connected metabolites that are the most difficult to negatively select. A further prediction derived from this situation is that the activity of many metabolic enzymes in vivo will be lower as the activity estimated in in vitro kinetic experiments. One illustrative example is TPI, a central enzyme in glycolysis. In vitro considered a 'prefect enzyme' limited only by the diffusion rates of its substrates, it has been suggested that TPI is multiple times more active in cells than is needed for glycolysis[43]. In vivo however, a reduction of TPI activity by just 20% is sufficient to affect yeast metabolism[44].

If enzyme inhibition emerges for structural reasons, it follows that structural similarity could form a basis for the evolutionary origin of regulatory mechanisms. An example is identified in the inhibition of fructose-1,6-bisphosphatase by fructose-2,6-bisphosphate, a regulatory metabolite in eukaryotic cells. Escherichia coli, like most bacteria, do not possess fructose-2,6-bisphosphate (F26BP), so the E. coli enzyme can't have evolved to be regulated by the compound. Yet, F26BP, structurally similar to fructose-1,6-bisphosphate does inhibit E. coli fructose-1,6-bisphosphatase in vitro. In eukaryotic cell that use F26BP for regulation, its binding affinity is increased, especially in the presence of AMP, a non-competitive inhibitor for fructose-1,6-bisphosphatase[8,45,46].

Structurally driven enzyme inhibition, in particular its high frequency in affecting virtually all enzymes, might contribute to difficulties in designing synthetic metabolic pathways[47]. To illustrate the magnitude of the problem, we have conducted a computational Gedanken experiment and expanded the human metabolic network by the reactions of the E. coli reconstruction (iJO1366)[48]. The newly introduced 112 bacterial metabolites would serve as inhibitors for 494 (80%) of the human reactions. Vice versa, 313 human metabolites are annotated inhibitors for 289 E. coli specific biochemical reactions. The network expansion would hence both inhibit the existing as well as the newly introduced enzymes. Support for the relevance comes from synthetic biology, where engineered biosynthetic pathways are regularly of low activity, and in many cases it is required to metabolically evolve the host so that the product is formed[47,49,50]. A classic example was to render yeast an efficient producer of the anti-malaria drug artemisinic acid, which took more than a century of single-person's work to achieve[51]. Localizing newly introduced metabolic pathways to organelles increases metabolic efficiency, a strategy that shields enzymes from their inhibitors[47,52]. In this context, the genome-scale importance of enzyme activators is not understood, and warrants a systematic investigation. According to the data as stored in BRENDA, non-catalytic enzymatic activators are far less frequent as enzymatic inhibitors.

Several mechanisms reduce the problem of enzyme inhibition by separating biochemical reactions from each other. Metabolic oscillations for instance, as observed within central carbon metabolism, separate reactions by time. In yeast, the occurrence of metabolic oscillations has been associated with the separation of oxidative from fermentative metabolism which at any given time reduces the number of active reactions[53]. Second, predominant in microbial communities but also found in eukaryotic cells, cooperative exchange of metabolites enables cells to specialize in metabolism. As a consequence, in each cooperating communal cell, only subsets of the metabolic reactions need to be active[54]. We speculate that reducing the problem of enzyme inhibition could be one contributing factor

that adds to the success of metabolite exchange interactions in microbial communities. Studying eukaryotic metabolism, we find the subcellular distribution of metabolic reactions in organelles is influenced by the presence of inhibitors. There are several benefits of cellular compartmentalization including separation of metabolite from competing reactions[55] and managing toxic intermediates from inhibiting growth[56]. Our data show that metabolism exploit subcellular compartmentalization also to reduce enzyme inhibition. Despite inhibitors and structurally similar metabolites enrich in the neighbourhood of a metabolic enzyme (Fig. 4), the distribution of enzymes prevents such an enrichment in all organelles (Fig. 5). Overall, compartmentalization in the human network reduces the number of inhibitory metabolite–enzyme interactions by up to half, and appears hence to be a highly effective global counteraction against metabolic enzyme inhibition.

In summary, we have constructed a global network of inhibitory metabolite–enzyme interactions by computationally merging detailed enzymological information with a genome-scale human metabolic network reconstruction. The obtained metabolite–enzyme inhibitor network pictures general principles that underlie enzyme inhibition as obtained on a cross-species basis. We find that thousands of inhibitory interactions render, in essence, every biochemical process sensitive to metabolic inhibition. Most inhibitory interactions emerge as consequence of a finite structural diversity that exists within the central parts of the metabolome. Structurally driven enzyme inhibition is mostly of competitive, but partially also of allosteric nature. We exemplify the latter by detailed enzymological and structural analysis of L-lactate dehydrogenase in complex with two allosteric inhibitors, whose inhibitory properties are explained by their structural similarity to its typical allosteric regulator, pyruvate. Finally, enzyme inhibition emerges mostly in the metabolic neighbourhood of the inhibited enzyme, is dominated by the most conserved and important metabolites, which specifically interact with particular enzyme classes dependent on their chemical nature. This situation is associated with compartmentalization in the eukaryotic cell that reduce metabolic enzyme inhibition up to half and specifically, by preventing an enrichment of enzyme inhibition within the metabolic neighbourhood of the eukaryotic organelles. While metabolic enzyme inhibition is key for the self-regulation of metabolism, its primary cause is hence the chemical structure of the metabolome. Many non-catalytic metabolite–enzyme interactions mount to a constraint for the function of cellular metabolic network, so that the cells evolved mechanisms to limit the number of unwanted and inhibitory metabolite–enzyme interactions.

## Methods

### Reconstruction of an enzyme-inhibition network on the genomic scale.
Curation and mapping strategy: An enzyme-inhibition network was constructed by mapping enzyme inhibitory information curated from experimental data collected in the BRaunschweig ENzyme DAtabase (BRENDA)[16] onto the human genome-scale metabolic model (Recon2)[17]. For every enzymatic reaction of the Recon2 model (in total 747 distinct EC numbers), a list of all inhibitors (in total 30,107 in which an inhibitor with multiple names appear multiple times; redundancy was removed by curation) were extracted from the BRENDA database. These compounds were assigned unique ID accounting to the Kyoto Encyclopedia of Genes and Genomes (KEGG)[57], human metabolome database (HMDB)[22] on the basis of entries from KEGG and HMDB databases by using merging algorithms[58] (http://cts.fiehnlab.ucdavis.edu/). Unique metabolite IDs (KEGG, HMDB) were then assigned to Recon2 metabolites, and mapped with the list of inhibitors. In total, 682 compounds, out of 30,107 BRENDA inhibitors, were successfully mapped to the Recon2 metabolites. These 682 metabolites inhibit 621 enzymatic reactions as contained in the Recon2 model. In the enzyme-inhibition network, a metabolite and an enzyme node were connected if a metabolite is an inhibitor for the enzyme, constituting in total 5,989 edges in the network. Information related to different kind of inhibitions including competitive, uncompetitive or noncompetitive for inhibitors of the enzyme-inhibition network was extracted from the information stored in BRENDA. A similar approach was followed for 289 E. coli-specific enzymes (according to iJO1366 metabolic reconstruction[48]). A list of

6,028 inhibitors was extracted from BRENDA for 289 E. coli-specific enzymes. These inhibitors were assigned to KEGG IDs and mapped with metabolites from E. coli model. iJO1366 model has in total 1,138 compounds, of which 851 compounds have KEGG ID, and 450 compounds were inhibitors.

### Classes of enzymes and inhibitors.
Classification of 621 enzymes and 628 inhibitors was conducted following KEGG nomenclature[57] for enzymes and HMDB for small molecules[22]. Inhibitors are classified among total 17 HMDB Superclasses (Supplementary Table 2). Three superclasses including (1) Aromatic Heteropolycyclic Compounds, (2) Aromatic Heteromonocyclic Compounds and (3) Aromatic Homomonocyclic Compounds were merged to one group named (i) Aromatic Cyclic Compounds. Additionally, eight other superclasses with a low number of metabolites each ((1) Alkaloids and Derivatives, (2) Aliphatic Homomonocyclic Compounds, (3) Aliphatic Heteropolycyclic Compounds, (4) Benzenoids, (5) Homogeneous Non-metal Compounds, (6) Organophosphorus Compounds, (7) Organooxygen Compounds, and (8) Aliphatic Heteromonocyclic Compounds) were merged to a single group named (ii) Others compounds. The other Superclasses were (iii) Nucleosides, Nucleotides, and Analogues, (iv) Organic Acids and Derivatives, (v) Amino Acids, Peptides and Analogues, (vi) Aliphatic Acyclic Compounds, (vii) Carbohydrates and Carbohydrate Conjugates, and (viii) Lipids. HMDB database entry is according to Version 3.5. It is however also important to note that there is inconsistencies in taxonomical classification of compounds with respect to different databases such as HMDB[22], KEGG[57], ChEBI[59] or LipidMaps[60]. Although HMDB compounds taxonomy uses thousands of chemical rules to classify compounds in functional categories, there are still some issues in assigning proper group to different compounds. For example, amino acids which are also aromatic compounds are placed in Amino Acids, Peptides and Analogues superclass, so that redundancy is prevented.

### Enrichment of class of inhibitors inhibiting classes of enzymes.
Inhibitors were counted for each class of compounds, and inhibitory interactions for each class of enzymes. The 'phyper' function in R was used for Hypergeometric test. $P$ value 0.05 was used as significance threshold. False discovery rate (FDR) for inhibition of a class of enzymes by a class of inhibitors was calculated as follows. First, inhibitors were randomly assigned to each inhibitor class, and this process was repeated for a hundred times. Then, an enrichment analysis for inhibition of a class of enzymes was performed on randomly assigned inhibitor classes. Finally, if $P$ value of inhibition was better for a random group of inhibitors than the original inhibitor class then the counter was increased. FDR = counter/100.

### Compound similarity.
Out of 682 inhibitors of the enzyme-inhibition network, structure-data files (SDF) were available for 638 metabolites in the PubChem database. Pairwise compound comparisons for all pairs of inhibitors were performed using with PubChem fingerprints (fpSim)[30], as well as other measures of similarity such as Drug molecule similarity derived by molecular fingerprints (calcDrugFPSim[31]), Maximum common substructure (Tanimoto Coefficient, Overlap Coefficient[32]) and Pairwise compound comparisons with atom pairs (cmp.similarity[30]).

### Metabolite structural similarity calculations.
To examine similarity between an inhibitor and substrates, a list of all substrates of the enzyme was extracted from the network reconstruction[17]. Then, for each inhibitor of this enzyme, the pairwise similarity to each of these enzyme metabolites was calculated, and only the best match considered as the similarity value. Further, to calculate similarity between inhibitor and any random compound, a list of random compounds of equal number as the enzymes metabolites was selected randomly from the list of all inhibitory metabolites. This process was repeated 50 times for both inhibitors and random metabolites. For example, a metabolite is an inhibitor for 5 enzymes (E1, E2, E3, E4, E5), that have 2, 3, 3, 2, and 4 substrates each respectively, constituting in total a list of 14 substrates connected to the inhibitor. First, the similarity between the inhibitor and all 14 substrate metabolites is calculated. Then, 14 metabolites are randomly selected from the inhibitor network and their structural similarity compared with the inhibitor mapped to the enzyme. This random sampling is repeated 50 times, and eventually, 50 times the similarity score of the most similar random compound is plotted against the similarity score of the most similar enzyme metabolite.

### Within and between metabolic pathway similarity.
Pathway to pathway similarity was calculated by extracting a list of pathways in accordance to the Recon2 model[17]. Then, for each metabolite the score between all compounds of pathways was pairwisely calculated, the shared compounds (that is, ATP, NADH, and so on), as well as metabolites that are not inhibitors in the network, were excluded. The similarity between two pathways is then expressed as the average score of all pairs. On the basis of number of inhibitions from one metabolic pathway to another, excluding the common metabolites, $P$ values were calculated for every pair of metabolic pathways using hypergeometric test. Inhibition from one pathway to the other considered significant when $P$ value < 0.01.

**Metabolite and enzyme essentiality.** For every inhibitor, a list of genes directly involved in its metabolism was extracted from the reconstruction[17]. If any gene in the list is essential according to cell-specific essential genes[61], then the metabolite was considered essential. Following the similar strategy for enzymes, a list of all genes associated with an EC number was used. If any gene in the list is essential according to cell-specific essential genes[61], the enzyme was considered essential.

**Compartment–compartment inhibition.** In the Recon2 model, there are in total seven compartments including Cytoplasm, Golgi apparatus, Lysosome, Mitochondria, Nucleus, ER and Peroxisome[17]. Enzymes and inhibitory interactions were assigned to these compartments according to the localization of its participating pathways. A $7 \times 7$ numerical matrix of metabolites inhibiting compartmentalized enzymes was assembled and used to define, for every pairwise comparison between compartments, trans-compartment inhibiting enzymes. Metabolites shared between compartments were excluded as indicated. For self-inhibition within a compartment, compartment-specific inhibitors are considered. These counts were used to calculate significance of inhibition between a pair of compartments using hypergeometric test. $P$ value $< 0.05$ was used as significance criterion to construct the compartment–compartment-specific inhibition network. Additionally, a FDR between a pair of compartments was calculated. First, for each compartment, the same number of compounds was randomly assigned, and this process was repeated for a hundred times. Then, an enrichment analysis for inhibition within and between a pair of compartments was performed following the same strategy as for the randomly assigned compounds. Finally, if the $P$ value of inhibition between a pair of compartments is better for a random set of inhibitors compared to the original inhibitors then the counter was increased. FDR = counter/100.

**Simulating the effect of a random compartmentalization.** To examine the overall impact of compartmentalization on inhibition, a number of random compartments ('rc', between 2 and 50) were created. First, for each metabolite 'm' and enzyme 'e' of the inhibition network (Fig. 1b), the number of compartments 'n' (between 1 and 'rc') in which 'm' and 'e' are allowed to participate was randomly selected following a power-law distribution ('function(x, a = 0.5, b = 1) a*b^a/x^(a + 1)'). Then, 'm' and 'e' was placed in 'n' random compartments. Finally compared to the total 5,989 interactions of the inhibition network (Fig. 1b), the total number of inhibitory interaction removed due to 'rc' compartments was counted. The entire process was repeated for 50 times.

**Selection against unwanted inhibition.** To examine whether the selection of unwanted inhibitions within compartments affected competitive and allosteric inhibition modes differentially, we used the set of 'gold standard' examples for which the type of inhibition is known with the greatest reliability (Fig. 2a). We calculated a ratio that compares the occurrence of competitive versus allosteric inhibition in a compartment-dependent manner. A ratio of substrate-inhibitor (S-I) pairs within the same compartment, versus the S-I pair within the different compartment, is more pronounced for competitive inhibition compared to allosteric inhibition (Fig. 5e).

**Kinetic characterization of L-LDH from rabbit muscle.** The activity of L-LDH was measured spectrophotometrically following the NADH oxidation at 340 nm at 37 °C in a 50 mM buffer $Na_2HPO_4$ pH 7.4 (final volume 150 µl). The reaction mixture contained 1 mM NADH, 50 ng of L-lactate dehydrogenase from rabbit muscle (Roche) and pyruvate was added to start the reaction. For $Km$ determinations, a saturation curve of pyruvate was generated by varying the concentrations from 0 to 3 mM at saturating concentrations of NADH (1 mM). For the saturation curve for NADH, this substrate was varied from 0 to 1 mM at saturating concentrations of Pyruvate (3 mM). For $Ki$ determinations, similar experiments for $Km$ determinations were done in the presence of different concentrations of oxaloacetate (0–7.5 mM). For the $Ki$ determination of oxaloacetate versus pyruvate, the reaction mixture containing NADH, L-LDH and oxaloacetate was incubated for 1 min and then the reaction was started by adding pyruvate. All the substrates and inhibitor were prepared fresh and the oxaloacetate stock solution was kept on ice maximum for 2 h. The kinetic data was analysed using the software Origin 9.0.

**Crystallization of L-LDH.** The L-lactate dehydrogenase from rabbit muscle (Roche) was buffer exchanged into HEPES 40 mM pH 7.5 using a centrifugal concentrator (Amicon Ultra-4 Centrifugal Filter Unit with Ultracel-30 membrane) and then adjusted to a final concentration of 5 mg ml$^{-1}$. For the L-LDH co-crystallized with NADH, 5 mM of the cofactor was mixed with the protein before the crystallization experiments. The crystals were grown using the vapour diffusion method at 20 °C in a solution containing 10% wt/v PEG 3350, 50 mM Bis-tris propane pH 6.5 and 100 mM NaF. The crystals appeared after 24 h reaching their maximum size after 4 days. The crystals were cryoprotected by adding the reservoir liquor supplemented with 20% w/v PEG 400 and 5 mM oxaloacetate before flash freezing in liquid nitrogen. In the case of the L-LDH co-crystallized with malonate, the protein was buffer exchanged as mentioned above. The crystals were grown in a buffer containing 0.1 M sodium malonate and 10% w/v PEG 3350. The crystals were cryoprotected by adding the reservoir liquor supplemented with 20% w/v PEG 400 and 2.5 mM oxaloacetate and then frozen immediately in liquid nitrogen.

**Crystallization data collection and structure analysis.** Data for the L-LDH crystals with malonate were collected at Diamond Light source on station I04-1, and data for the L-LDH crystals with NADH and oxaloacetate were collected on station I03. Both data sets were collected at 100 K. The structures were solved by molecular replacement (PHASER) using coordinates for rabbit muscle L-LDH in which all ligands and water molecules were removed (PDB code 3H3F (ref. 62)). The structures were refined using PHENIX and ligands built into density using COOT. The diffraction data were processed avoiding images with radiation damage. The refinement and crystallographic data are summarized in Table 1.

**Additional software tools.** SOAP from BRENDA (http://www.brenda-enzymes.org/soap.php)[16] was used to fetch inhibitors for every enzyme. The displ, dislnorm, dispois and disexp functions from R package poweRlaw[63] were used to fit power-law, log-normal, Poisson and exponential distributions respectively for the nodes of enzyme-inhibition network as well as for a random network which was constructed by function erdos.renyi.game from R package igraph[64]. Shortest path length for metabolites and enzymes in the enzyme-inhibition network was calculated by function shortest.paths. Word cloud of inhibitors was made by wordcloud library in R. Alignment of substrate and inhibitor in active site of enzymes was done by PyMOL tool. The figures were prepared using Adobe Illustrator. Function cor.test in R was used for Pearson's product–moment correlation. Function t.test in R was used for Welch two sample $t$-test.

**Data availability.** The data that support the findings of this study are available from the corresponding author on reasonable request. The coordinates and structure factors for the malonate and oxaloacetate with NADH complexes with rabbit muscle L-Lactate dehydrogenase have been deposited with the PDB accession codes 5NQB and 5NQQ, respectively. More detailed information about the Inhibition Network, Inhibitors, Enzymes, Competitive inhibition, Noncompetitive inhibition, Uncompetitive inhibition and Pathways are available in Supplementary Data 1.

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

## Acknowledgements

We thank Elahe Radmaneshfar and Gianmarco Meo for help with data analysis and experiments, Emile Van Schaftingen (UC de Louvain, Belgium), Kevin Brindle (CRUK Cambridge Research Institute), Jules Griffin (University of Cambridge) and Kiran Patil (EMBL Heidelberg) for critical reading and providing profound insights into the enzyme inhibition. We thank the staff of Diamond Light source for use of facilities (stations I04-1 and I03). We acknowledge support by the Francis Crick Institute which receives its core funding from Cancer Research UK (FC001134), the UK Medical Research Council (FC001134), and the Wellcome Trust (FC001134), the European Research Council (ERC Stg 260809 to MR) and Wellcome Trust (RG 093735/Z/10/Z, to MR), to the Consejo Nacional de Ciencia y Tecnología Mexico postdoctoral fellowship 232510 to V.O.-S. A.Z. is EMBO fellow (ALTF-969 2014) which is co funded by the European Commission (LTFCOFUND2013, GA-2013-609409) support from Marie Curie Actions. M.A.K. is supported by an Erwin Schrödinger postdoctoral fellowship (FWF, Austria, J3341).

## Author contributions

M.R. developed the concept and designed the study, M.T.A., V.O.-S., A.S., M.A.K., A.Z. developed the methodology and conducted the computational analysis, V.O.-S. and B.F.L. generated the crystallographic structures and conducted enzymatic assays. M.R. wrote the first draft of the manuscript, all authors contributed in finalizing the manuscript.

## Additional information

**Competing interests:** The authors declare no competing financial interests.

