## [Peer review file · Nature Communications]

Reviewers' comments:

Reviewer #1 (Remarks to the Author):

The paper of Alam et al. addresses a largely neglected issue of metabolic organization: how does the map of metabolite - enzyme inhibition interaction look like? This is an original manuscript with many important findings: i) provides the first large-scale network of enzyme inhibitions based on compiling decades of enzymology data, ii) it shows that competitive inhibition is the major mode of inhibition and there are plenty of metabolites that can act as inhibitors owing to chemical similarity to native substrates, iii) even allosteric inhibitions are often caused by chemical similarity and this is demonstrated experimentally in a case study, iv) probably the most important finding is that compartmentalization has evolved to minimize inhibitions.

Overall, I found most analyses convincing, however, I had few technical issues and would like to see some additional analyses to support the claims.

Major points:

- Reconstruction: Recon2 has 1789 enzymes, but the authors mention 747 enzymes that form Recon 2. Does it refer to distinct EC numbers? If so, this should be clearly explained.
- The authors claim that the metabolite inhibition network follows a scale-free distribution. Some statistical evidence against alternative distributions should be presented to support this claim (e.g. see Science 335: 665-666 for examples). Btw, Fig 1c would be more informative on log-log scale. Most importantly, it is not fully clear what biological / chemical message a scale-free distribution entails?
- In two parts of the manuscript (enrichment of class of inhibitors and enrichment of inhibitor within versus between compartments), enrichments are calculated based on a hypergeometric test. As the connectivity distribution in the underlying network is far from normal, it would be more appropriate to calculate enrichments based on randomizing the inhibition network (with keeping the same in and outdegrees) and comparing it to the real network.
- Compartmentalization and selection against unwanted inhibition: this should be most pronounced for competitive inhibitors. Allosteric inhibition might be more easily removed by altering the protein sequence and hence one may expect a less pronounced compartmentalization effect there. This could be tested by repeating the within vs between compartment inhibitor enrichment analysis separately on the two subsets of inhibitions.
- The authors found that the more likely that a metabolite is an inhibitor, the more likely it participates in an enzymatic reaction that is essential for the cell. I wonder if this result is not a by-product of the definition of essential metabolites (i.e. a metabolite is considered essential if at least one gene directly involved in its metabolism is an essential gene) and the fact that highly connected metabolites also have more inhibitory interactions? In addition, it may be worth checking whether this pattern is not caused by central metabolites being more likely to be essential and showing more similarity with each other.
- The authors hypothesize that 'Metabolism in eukaryotic cells is compartmentalized in organelles, which by separating reactions, may reduce the number of enzyme-inhibitor interactions for eukaryotic metabolism.' I wonder if this idea and some case studies have been proposed by earlier works? If so, these references should be added.

Minor points:

- The text would benefit from editing (e.g. shortening some difficult-to-follow sentences).
- As I understand, the authors used information on orthologous enzymes from other species as well when reconstructing the inhibition network. This is very reasonable, however, they refer to this process as getting information on 'human paralogues' (p 6 and p 16.), which incorrectly implies that information was retrieved from a duplicate copy in the human genome, not a homolog from another species (ortholog).
- Fig 1f: edge width should denote not the p-value of inhibition but rather the enrichment of inhibition (i.e. effect size)
- It might be worth mentioning that competitive inhibition is easier to detect and hence could bias our estimate on the prevalence of the different modes of inhibitions.
- Some statistical associations are reported in the text without explicitly showing the statistics (e.g. R and p-values) in the text (e.g. page 13)
- p 18: "Most importantly, enzyme inhibition is predominantly an emergent property of the structural makeup of the metabolome." -- I had difficulty to interpret this statement.
- inhibitory reaction -> inhibitory interaction seems as a more appropriate term as inhibition does not involve chemical interconversion of metabolites
- Figure 5E: some more description is needed in the figure legend on how to interpret this plot.

Reviewer #2 (Remarks to the Author):

NCOMMS-16-25778 "Enzyme inhibitors assemble in a global metabolite-enzyme interaction network that broadly constraints metabolism" by Alam et al. In this m/s the authors report an thorough analysis of the BRENDA database, showing a high level of (potential) inhibition of enzymatic activity by metabolites, most of which is of competitive nature and can be explained by chemical similarity. The authors also analyze the effect of cellular compartmentalization. The m/s covers an interesting topic and it is clearly written, with well-structured figures. Overall, I am positive about its potential publication in Nature Communications. However, there are a few points that should be clarified. Please, find below some specific comments that might help the authors improving the quality and robustness of their results.

After analyzing BRENDA, the authors find a surprisingly high number of metabolite-induced enzymatic inhibition. They find that most of these inhibitory interactions are competitive, and can be explained by chemical similarity between the natural substrates of the enzymes and other related metabolites. They also find that, not surprisingly, the most similar compounds are those belonging to close metabolic reactions within the same pathways (i.e. Fig S3b) and, obviously, most metabolic routes are contained within the same subcellular compartment (particularly if the metabolites cannot freely diffuse between different localizations). However, when the authors analyze the enzymes in each compartment separately, they find that most of the intra-compartment inhibitory interactions are lost and, instead, enrichments are found between different compartments (i.e. Fig 5f). In my opinion, this is very counter-intuitive and questions the results reported in preceding sections of the m/s.

Additionally, the authors are assuming that substrates/products of enzymes confined to a particular compartment cannot freely diffuse, which is certainly not the case for many of them. Thus, I guess, that the biological reality will be somewhere between the two scenarios analyzed. I would suggest them to clarify these points and re-run ALL their analyses considering the two possibilities, and see whether the results change or hold (i.e. are intra-compartment inhibitions

also competitive and driven by structural similarity?).

The authors also find a significant correlation between the number of inhibitory reactions and the number in which a given metabolite participates. However, I suspect that this correlation is the result of the above mentioned chemical similarity between compounds in adjacent reactions. If the authors want to make the point, they should try to detach both effects.

Finally, the authors discuss a potential lack of coherence between reported enzymatic constants (K_m and K_i) and the real inhibitory effect of metabolites *in vivo*, and they speculate that strong inhibitory constants might become irrelevant if the enzymes and metabolites are in different compartments and vice versa. I agree with their observation that quantitative data is highly heterogeneous and incomplete, however, it would be interesting to see if there is any correlation between K_m/K_i reported constants within or between compartments and well documented cases of *in vivo* inhibition, and see if any sort of "biologically relevant" inhibition threshold could be defined.

Reviewer #3 (Remarks to the Author):

In this study, Alam et al. use the vast information accumulated in the last century on enzyme inhibition to draw a semi-global network of metabolites and the enzymes they inhibit. The authors analyze this inhibition network and uncover multiple interesting phenomena that directly relates to our understanding of the function and evolution of metabolism. The work described in this study is innovative, provocative and of interest for very broad range of readers. However, it does suffer from several problems, which I list below. Do these problems prohibit the publication of the study? Honestly, my mind is split here and I will let the editor decide.

Major concerns

1. Multiple measurements biases might affect, and ultimately be responsible for, many of the trends noticed by the authors. While the authors are aware of it, and refer to this issue in multiple places, the explanations they provide for justify the results are not fully convincing.

To give some examples:

"Each enzyme class is comparably well covered by inhibitor information, ruling out that enzymological data does bias against a certain type of enzyme" – the fact that all enzyme classes are represented does not mean that there is no bias towards specific types of enzymes – some might still be measured many times while others only few times.

"By far the largest total number of inhibitions are mediated by the HMDB superclass Nucleosides, Nucleotides, and Analogues ..." – this could be the results of measurement bias, as ATP and analogs were probably tested much more than any other type of compounds.

"Despite some metabolites, like for instance ATP, have certainly been more studied than other metabolites, they act as typical inhibitors on a subset of enzyme classes..." – again, can't it be that for specific enzyme classes, like ligases, which utilize nucleotides as substrates, "Nucleosides, Nucleotides, and Analogues" were tested as inhibitors more frequently than for other enzyme classes?

"...and that clear patterns emerge, despite the potential coverage bias, caused by the fact that the inhibition network was assembled from small scale enzymological data." – I still don't understand how the coverage bias is factored out here.

"the closer the metabolite is in the metabolic network to the substrate of the enzyme (within the same metabolic pathway), the more likely it has been discovered as inhibitor for the particular enzyme by a study listed in BRENDA" – this could be a classic measurement bias; enzymologist characterizing an enzyme were biased to test other metabolites in the pathway for inhibition.

"we found that the central metabolic pathways glycolysis, gluconeogenesis and nucleotide interconversions, were the most inhibited and inhibiting metabolic pathways" – could be another measurement bias. The authors try to explain, in the following lines, why this is no so, but I failed

to understand the explanation. [for example, what is an "inhibitory reaction"? Could be either an inhibitory compound or an inhibited reaction...]

"the large amount of enzyme data covered the metabolic network reconstruction to the extent that we find no selective bias against a certain enzyme class or reaction mechanism, even though some have been clearly more studied than others" – I again fail to understand this logic. The fact that there exist a lot of data does not mean that there are also significant measurement biases.

"as a rough proxy of metabolite's market properties, we plotted the price of the chemicals in the 2016 Sigma-Aldrich catalogue against the number of associated inhibitory reactions, and found no correlation" – this is far from being convincing.

"The nature of this network shows that despite the coverage bias that exists within the existing enzymological data, a clear preference between metabolic inhibitor and the affected enzyme classes is pictured." – again I fail to see the argument here – see above.

2. Why did the authors ignore the list of activator compounds? These are also given in BRENDA and are equal in importance. In a sense, inhibitors and activators are the same thing with a reverse sign, and to get a full picture of the inhibition/activation network, they should be included. Of course, considering compounds that bind to the same active site as the substrate, only inhibitors make sense, but for allosteric effects, inhibitors and activators serve the same basic role.

The authors say: "Due to the presence of inhibitory metabolites for virtually each enzymatic reaction, the activity of many metabolic enzymes in vivo would be lower than expected from in vitro kinetics" – however, this is not true when activators are considered. Indeed, the measured *k_{cat}* of many enzymes (e.g., *E. coli*'s SerA) is far too low to explain their in vivo activity and hence there must be some activators that significantly enhance their activity in vivo.

Minor concerns

1. From previous experience I had with the BRENDA database I found that ~20% of the kinetic data reports are erroneous, i.e., do not fit the data reported in the corresponding papers, due to mistakes in copying or unit mismatch. While I think that the inhibition data is more accurate it is still worthwhile reporting the errors found and how a general analysis could deal with such errors.

2. The authors claim that enzyme inhibition is typically not specific to a given organism and hence it is justified to integrate data from other organisms to create the human inhibitory network. I'm not convinced that this is indeed the case. For example, multiple enzymes that are represented by the same EC number, actually correspond to different catalytic mechanisms, or 'Classes', such that we (human) harbor an enzyme from one class and another organism harbors an enzyme from a different class – both are completely different in mechanism and inhibition.

3. The inhibitors were classified to different groups. However, the presence of phosphate was not taken into account. For example, I think that differentiating between "Carbohydrates" and "Phosphorylated-Carbohydrates" might be highly interesting as it is well-known that protein binding to phosphate groups has a special significance.

4. "33% of noncompetitive and 31% uncompetitive did however possess significant structural similarity to the enzyme's substrate (Fig 2b)" – I find this statement a bit problematic as the distributions in Figure 2b seem rather spread over all the similarity range. Hence, it seems that the structural similarity plays no actual role in allosteric inhibitors, where some, just by mere chance, are similar to the substrate. I do not dispute the results but simply the way they are presented: instead of saying that 2/3 are not similar and 1/3 is similar, I would say that it seems that similarity plays little role in allosteric inhibitors.

5. "...and use the human metabolic network (recon2), one of the most comprehensive metabolic network reconstructions, as a basis of our analysis." – here I wonder why the authors did not try to further use *E. coli* or yeast to strengthen their findings – there is vast data on the enzymes of

both of these microbes.

6. "It follows that a majority of enzyme inhibition by cell's metabolites did not evolve to regulate metabolism. Instead, enzyme inhibition emerges for structural reasons, which implies that many inhibitory interactions have negative impact on the functionality of metabolic network." – this is an interesting assumption to which I tend to agree. Still I do not think that the authors "proved" it and hence I would state it as a mere assumption.

7. "one has to take into consideration that the metabolites which are part of the modern metabolic pathways are the ones that prevailed during evolution, while strong inhibitors like the aforementioned 4-phospho-erythronate and 2-phospho-L-lactate had to be removed from the system in order for cells to survive" – I would be very careful here as there are highly reactive/toxic/inhibitory compounds which are used as metabolites by some organisms and at some conditions; for example, formaldehyde, methyglyoxal, and hydroxyserine.

Reviewer #4 (Remarks to the Author):

The manuscript by Alam et al. attempts to provide global analysis of the metabolome and its constrains. Understanding cellular networks and their regulations, interconnectivity and constrains is key to physiology and biology. The authors focus on non-catalytic metabolite enzyme interactions, which may play a role in major regulatory functions. This is very interesting manuscript but quite uneven.

The authors merged enzyme knowledge accumulated over the past 100 years with genome-based metabolite enzyme-inhibitor networks. To validate their hypothesis, they use published and new structural studies of allosterically inhibited L-lactate dehydrogenase. Unfortunately they were unable to compare K_i/K_m for these enzymes and metabolites.

Here I comment on the quality of the crystal structures reported in the paper. The crystal structure of rabbit muscle L-LDH was determined as a complex with oxaloacetate and NADH cofactor and with malonate. These two structures are high resolution and should be very similar but showed some different stats. It is quite difficult to judge just based on Table S4 because I do not have access to the data and PDB report. I do not question the validity of the structures (unfortunately no electron density is available), but there are questions on how these structures have been done. For example, the higher resolution structure has poorer rms on bond length and angles. Why? There are some inconsistent things with the data and refinement statistics. It appears that the lower resolution structure is "better" than the higher resolution structure. The higher resolution structure has significantly less observations (less than half) than the lower resolution structure. Why is that? The coverage is poorer for higher resolution structure as well as I/σ . At the same time more unique reflections are used for refinement, which is quite strange. Even stranger is the fact that these structures seem to be quite an important part of the observations but they are not discussed in "Discussion" section. Moreover, there are a number of crystal structures of human and rabbit LDHs but they are not mentioned at all.

In general this is very interesting manuscript but it is quite sloppy and there are a number of minor issues with arguments, text and English.

One example is: "studied by X-ray diffraction" rather than "studied by X-ray crystallography"

Point-to-point response to all reviewer's comments and suggestions

Reviewer 1

The paper of Alam et al. addresses a largely neglected issue of metabolic organization: how does the map of metabolite - enzyme inhibition (P-value $4.8e-34$ interaction look like? This is an original manuscript with many important findings: i) provides the first large-scale network of enzyme inhibitions based on compiling decades of enzymology data, ii) it shows that competitive inhibition is the major mode of inhibition and there are plenty of metabolites that can act as inhibitors owing to chemical similarity to native substrates, iii) even allosteric inhibitions are often caused by chemical similarity and this is demonstrated

experimentally in a case study, iv) probably the most important finding is that compartmentalization has evolved to minimize inhibitions.

Overall, I found most analyses convincing, however, I had few technical issues and would like to see some additional analyses to support the claims.

We thank the Reviewer for the summary and supportive general comments, and particularly for some valuable suggestions that we enjoyed incorporating and that strengthened the paper, as explained below:

(1) - Reconstruction: Recon2 has 1789 enzymes, but the authors mention 747 enzymes that form Recon 2. Does it refer to distinct EC numbers? If so, this should be clearly explained.

We apologize that this was unclear and that we have misleadingly used the word 'enzyme' in several instances where this was not a precise enough term. The reviewer is correct, we have used the 747 distinct EC classifiers that make-up the recon2 metabolic reconstruction as the basis for the inhibition network (in the curated version this number reduced to 621 as not all enzymatic reactions have inhibitors associated that match the set criteria). We have now clarified this throughout the manuscript, and now use the word 'EC classifier', or '*distinct EC number*', or '*biochemical/enzymatic reaction*'; instead of just 'enzyme' in most positions, so that we are semantically precise.

(2) - The authors claim that the metabolite inhibition network follows a scale-free distribution. Some statistical evidence against alternative distributions should be presented to support this claim (e.g. see Science 335: 665-666 for examples). Btw, Fig 1c would be more informative on log-log scale. Most importantly, it is not fully clear what biological / chemical message a scale-free distribution entails?

We have expanded the statistical analyses as suggested by the Reviewer. Other than power-law distribution, we have fitted three alternative distributions including log-normal, Poisson and exponential distribution to the degree distribution of the inhibition network. The model comparisons were performed for power-law distribution against the three other distributions and the P-values were reported. Further, as suggested, the plot is now shown in log-scale (Figure 1c).

The results are as follows: For the inhibition network, the degree distribution follows a power-law and a log-normal distribution (P-value for comparing power-law distribution with log-normal, Poisson and exponential are 0.36, 8.7e-05, 0.077 respectively), whereas for a random network, the P-value for comparing power-law distribution with log-normal, Poisson and exponential are 0.42, 0.78, 0.44 respectively. For the inhibition

network, the fitted parameter gamma from a power-law distribution is 2.93, compared to that of a random network where gamma is 11.47, is a further indicator that the network is a scale-free network. We have revised the method and results section and figure legend in the manuscript to incorporate fittings of some other distribution functions.

The rationale for the inclusion of this analysis to the manuscript: We have presented several measures throughout the manuscript that show that the inhibition network resembles a non-random, typical, biological network. This is, in part, to destroy some Readers/Reviewer's concerns (i.e. Reviewer #3), that the network could be representing a random bias of the existing enzymological data. Demonstrating the nature of its connectivity is just one of these measures of course, and we agree that there is not much more to learn from this fact if looked at in separation. Finally, as considerable importance has been placed on power-law distributions for biological networks, any biological network description would feel incomplete without a topological analysis.

Response Figure 1: Refined statistics confirm the enzyme-inhibitor network to be a non-random biological network following a power-law distribution.

(3) - In two parts of the manuscript (enrichment of class of inhibitors and enrichment of inhibitor within versus between compartments), enrichments are calculated based on a hypergeometric test. As the connectivity distribution in the underlying network is far from normal, it would be more appropriate to calculate enrichments based on randomizing the inhibition network (with keeping the same in and outdegrees) and comparing it to the real network.

We thank the reviewer for an excellent suggestion. We have performed a new enrichment analysis according to randomly generated connections and repeated that one hundred times to calculate a false discovery rate (FDR) for the enrichment over the biological inhibitor network. Results were incorporated in the result section and FDR values were highlighted in respective figures (Figure 1f and Figure 5f, Supplementary table S4, S5). Additionally, the analysis approach was explained in the method section.

Figure 1f

Figure 5f

Response Figure 2: Revised networks, FDR corrected against random networks. (Figure 1f) The different inhibitor chemical classes show specificity about which enzyme class they inhibit. chemical. Size of the nodes is scaled according the number of inhibitor/enzymes within each class. The edge thickness is scaled according to the occurrence of significant inhibitory interactions between the inhibitor's superclass, and enzyme class. Nodes are connected if $P\text{-value} < 0.05$. FDR values are highlighted over

edges. (Figure 5f) Compartment-compartment specific inhibition network in which nodes represent the seven subcellular compartments. Edges are drawn if the number of inhibitor compounds is significant (P -value < 0.05). FDR values are highlighted over edges.

(4) - Compartmentalization and selection against unwanted inhibition: this should be most pronounced for competitive inhibitors. Allosteric inhibition might be more easily removed by altering the protein sequence and hence one may expect a less pronounced compartmentalization effect there. This could be tested by repeating the within vs between compartment inhibitor enrichment analysis separately on the two subsets of inhibitions.

The reviewer has made an interesting suggestion that we did like very much. A respective analysis was performed on the set of 'gold standard' examples for which we know the type of inhibition with reliability, and results and approach were incorporated in the manuscript (see Figure 5). The outcome is that compartmentalisation removes, in relative terms, more allosteric inhibitory interactions as it removes competitive inhibitory interactions.

Response Figure 3: The ratio of the substrate-inhibitor (S-I) pair within the same compartment, versus the S-I pair within the different compartment is more reduced for allosteric inhibition as it is for competitive inhibition

(5) - The authors found that the more likely that a metabolite is an inhibitor, the more likely it participates in an enzymatic reaction that is essential for the cell. I wonder if this result is not a by-product of the definition of essential metabolites (i.e. a metabolite is considered essential if at least one gene directly involved in its metabolism is an essential gene) and the fact that highly connected metabolites also have more inhibitory interactions? In addition, it may be worth checking whether this pattern is not caused by central metabolites being more likely to be essential and showing more similarity with each other.

We orient in our manuscript on experimental genetic data to define which enzyme is essential; and derive metabolite essentiality from these results (a metabolite is termed 'essential' if it's essential for an essential enzymatic reaction). The reviewer is right that we are not able to disentangle essentiality and connectivity. However, there is a biological reason for this. It's intuitive that some metabolites are essential *because* they are participating in a large number of biochemical reactions, while it's also intuitive that a metabolite that participates in only one or a few enzymatic reactions is less likely essential, and can be easier negatively selected from the metabolic system during evolution. As our paper then further shows, the most connected metabolites are more likely a competitive inhibitor, in essence, as there are more enzymes around that have evolved to bind to them.

We tested the suggestion of the Reviewer that essential metabolites might be overall more structurally similar amongst each other, compared to non-essential metabolites. The reviewer was right. We found this connection highly significant in an analysis in which we calculated the average compound similarity in and between essential metabolites and non-essential metabolites. The average compound similarity between essential – essential metabolites is significantly higher than the average compound similarity between all other cases, i.e. nonessential – nonessential, essential – nonessential, and between nonessential – essential compounds (new Figure S6). In other words, nonessential and secondary metabolites have more divergent structures, and at the same time less frequent competitive inhibitors as essential metabolites. We find this result intriguing, but the rationale is the following: Secondary and distal metabolites have more divergent structures as they are in the periphery of the metabolic network, and hence have undergone more chemical conversation as central metabolites. The text in results and discussion section has been updated accordingly, and a new figure (Figure S6) has been included in the manuscript.

Please Refer also to the response to Reviewer #2, however, one result clearly shows that while metabolite similarity is a driver of inhibition, the two properties are differently affected by evolution. This becomes apparent in the compartmentalisation analysis: While compartment-localized metabolites are structurally similar to one another, they are not enriched for inhibition as one might expect from the situation that prevails in the cytoplasm. Chemical structural similarity and being an inhibitor are hence related, but not the same, property. Further, ALL of the most frequent inhibitor are essential metabolites, but not all essential metabolites are frequent inhibitors. The two are hence clearly related, but in the end they remain separate properties.

Response Figure 4: Essential metabolites show a increased structural similarity to one another, compared to other metabolites in the metabolic network. This figure is in Supplemental information **Figure S6: Average compound similarity scores between essential - essential, nonessential - nonessential, essential - nonessential, and between nonessential - essential metabolites are shown in a (a) boxplot and in a (b) density plot. The average compound similarity between essential - essential metabolites is significantly higher than average compound similarity between all other cases, i.e. nonessential - nonessential, essential - nonessential, and between nonessential - essential compounds.**

(6) - The authors hypothesize that 'Metabolism in eukaryotic cells is compartmentalized in organelles, which by separating reactions, may reduce the number of enzyme-inhibitor interactions for eukaryotic metabolism.' I wonder if

this idea and some case studies have been proposed by earlier works? If so, these references should be added.

There is certainly several literature about the role of compartmentalisation, and some address the compartmentalisation of 'toxic' metabolites. We apologize that in previous version of the manuscript a general discussion about compartmentalization of metabolism has been omitted. We have expanded the discussion accordingly and added several literature citations concerning the general and metabolic nature of compartment formation. However, we have not spotted a significant manuscript that addressed the role of structural similarity between metabolites as consequence of the metabolic network structure and considers the emergence of enzymatic inhibition as a direct consequence, which is the key message of this paper.

Minor points:

- The text would benefit from editing (e.g. shortening some difficult-to-follow sentences).

We have made extensive changes to the text, and hope the manuscript is now much easier to read and to dissect. We acknowledge the manuscript remains dense, as it is rich in results, as we aim to support all statements made with an analysis (~50 figure panels).

- As I understand, the authors used information on orthologous enzymes from other species as well when reconstructing the inhibition network. This is very reasonable, however, they refer to this process as getting information on 'human paralogues' (p 6 and p 16.), which incorrectly implies that information was retrieved from a duplicate copy in the human genome, not a homolog from another species (ortholog).

We agree that our terminology was imprecise in several instances when using the term 'paralogue', and we have corrected it.

- Fig 1f: edge width should denote not the p-value of inhibition but rather the enrichment of inhibition (i.e. effect size)

This has now been corrected. Edge width now indicate "enrichment of inhibition"

- It might be worth mentioning that competitive inhibition is easier to detect and hence could bias our estimate on the prevalence of the different modes of inhibitions.

We entirely agree with the reviewer that theoretical prediction of which metabolite is a potential inhibitor is much easier to achieve for competitive inhibitors (as they bind to the same site or site near the catalytic site, AND are structurally similar; while structural similarity are less a condition for an allosteric inhibitor). In purely experimental studies however – where the mechanism is not known *a priori* and typically determined *after* a substance is found to be an inhibitor – we expect only a minor difference in detecting competitive versus allosteric inhibitors (both competitive and allosteric inhibitors can have low or high binding affinities, and are identified in the same experiments or screens, and as our analysis shows, both allosteric and competitive inhibitors are most likely found in the close neighborhood of an enzyme). As experimental studies make up (by far) most of the enzymological data, including ~virtually all studies published before the 2000's, the bias from computational inhibitor prediction is hence small. Indeed, for the vast majority of studies as collected in Brenda a the mechanism/type of inhibition is not known at all. In this set we score a similar average structural similarity between substrate and inhibitor, as we do for the gold-standard dataset where the reaction mechanism is known.

Although we consider the impact of this particular bias to be small, we agree it might exist, and included this potential caveat as suggested. Most importantly however, please also note that there is not a small but in fact a massive difference in the occurrence of competitive inhibition versus the allosteric inhibition types (Figure 2). Given the richness of our network that bases on ~200,000 publications, a hypothetical detection bias favouring the discovery of competitive over allosteric inhibitors in experiment would have need to be extreme in order to affect the conclusion that competitive inhibition is more frequent as compared to allostery would prove incorrect. Considering the 'gold-standard set', that reports 75% of inhibitions to be competitive, this would require most enzyme mechanisms to be wrongly reported by enzymologists and in fact be allosteric (a result which would be counter-intuitive).

- Some statistical associations are reported in the text without explicitly showing the statistics (e.g. R and p-values) in the text (e.g. page 13)

We thank the reviewer for the suggestions and have corrected these throughout the manuscript. For instance, in Figure 4c we have newly added a T-test to compare the significant difference between 'average compound similarity between different pathways' and 'average compound similarity between the same pathways' and the p-value 1.91e-14 was highlighted in the figure as well as mentioned in the main text. Additionally, P-value from Figure 4b was also highlighted in the main text.

- p 18: "Most importantly, enzyme inhibition is predominantly" -- I had difficulty to interpret this statement.

We apologize this statement was difficult to interpret. We have revised and extended this part extensively.

- inhibitory reaction -> inhibitory interaction seems as a more appropriate term as inhibition does not involve chemical interconversion of metabolites

We thank the reviewer for spotting this out and we have corrected it in many instances

- Figure 5E: some more description is needed in the figure legend on how to interpret this plot.

We thank the reviewer for pointing out that Figure 5e was difficult to understand. Due to the other revisions made, the referred figure has migrated to the Supplementary materials (Figure S7) and it has been replaced with a new figure generated in response to comment #4 of the reviewer. We also have explained the figure better by extending the figure legend:

Figure S7 (a) Average number of intra- and inter-compartmental inhibitory interactions per inhibitor. (b) Average number of intra- and inter-compartmental inhibitory interactions per inhibitor after removing the 16 'hub' inhibitors (from Fig 4k) which are participating in the metabolism of more than 5 compartments. (a) The total number of intra- and inter-compartmental inhibitory interactions after removing the 16 'hub' inhibitors (from Fig 4k) which are participating in the metabolism of more than 5 compartments.

Reviewer 2

NCOMMS-16-25778 "Enzyme inhibitors assemble in a global metabolite-enzyme interaction network that broadly constraints metabolism" by Alam et al. In this m/s the authors report an thorough analysis of the BRENDA database, showing a high level of (potential) inhibition of enzymatic activity by metabolites, most of which is of competitive nature and can be explained by chemical similarity. The authors also analyze the effect

of cellular compartmentalization. The m/s covers an interesting topic and it is clearly written, with well-structured figures. Overall, I am positive about its potential publication in Nature Communications. However, there are a few points that should be clarified. Please, find below some specific comments that might help the authors improving the quality and robustness of their results.

We thank the Reviewer for this very positive overall assessment of our manuscript.

(1) After analyzing BRENDA, the authors find a surprisingly high number of metabolite-induced enzymatic inhibition. They find that most of these inhibitory interactions are competitive, and can be explained by chemical similarity between the natural substrates of the enzymes and other related metabolites. They also find that, not surprisingly, the most similar compounds are those belonging to close metabolic reactions within the same pathways (i.e. Fig S3b) and, obviously, most metabolic routes are contained within the same subcellular compartment (particularly if the metabolites cannot freely diffuse between different localizations).

However, when the authors analyze the enzymes in each compartment separately, they find that most of the intra-compartment inhibitory interactions are lost and, instead, enrichments are found between different compartments (i.e. Fig 5f). In my opinion, this is very counter-intuitive and questions the results reported in preceding sections of the m/s.

We apologize if our writing was confusing on this point, and have re-worked the related paragraph extensively to avoid any misinterpretation. We did not claim or show at any point in the manuscript that “most of the intra-compartment inhibitory interactions are lost”. Indeed, Figure 5c,d shows that there are hundreds of intra-compartment inhibitory interactions left due to intra-compartment inhibitors (Figure 5b). But there is a statistically highly significant difference between the subcellular compartments and the random expectation or the cytoplasm, and this difference points to a selection pressure to distribute enzymes in the cell so that enzyme inhibition in the organelles is significantly reduced: As the reviewer states, due to metabolite structural similarity in adjacent metabolic reactions (Suppl Figure S4, S5), one would expect inhibitors to be enriched within the compartments. Our random compartmentalisation simulation, as well as our analysis of the distribution of inhibitors in the network space, as well as several other of our analyses, do confirm this expectation. However, the enzymatic reactions are distributed in a way that this enrichment does not occur in the organelles, intra-compartment inhibition is only enriched in the cytoplasm, but in none of the organelles. Instead, the control analysis of ‘inter-compartment inhibition’ scores significantly between most of the compartments, so it’s indeed the localisation in the

particular organelle that results in this lower degree of inhibition (Fig 5b). We have re-worked this part extensively so that the message will become clearer.

(2) Additionally, the authors are assuming that substrates/products of enzymes confined to a particular compartment cannot freely diffuse, which is certainly not the case for many of them. Thus, I guess, that the biological reality will be somewhere between the two scenarios analyzed. I would suggest them to clarify these points and re-run ALL their analyses considering the two possibilities, and see whether the results change or hold (i.e. are intra-compartment inhibitions also competitive and driven by structural similarity?).

We fully agree with the reviewer that one has to account for multiple localisations, and we apologise that from methods section it was apparently not sufficiently clear that we did already account for multiple localisations in all instances where compartmentalisation applies. We have re-worked this section for clarity. In our compartmentalization analysis (Figure 5a), we did take fully into account that a metabolite or an enzyme can be in more than one compartments according to their occurrence in the metabolic network reconstruction. In the random simulation, the number of compartments a metabolite or an enzyme is allowed to participate was randomly selected following a power-law distribution – just like in a typical biological system where degree distribution follows a power-law distribution. Indeed, we cannot ignore multiple localisation in this comparison, otherwise, we would violate the structure of the network: ignoring multiple localisation would restrict key molecules like ATP, NADPH to a single compartment, while the many enzymes that depend on these molecules would still be required across compartments.

However to address the reviewer' request for a hypothetical comparison without accepting multiple localisation (i.e. to quantify the importance of multiple localisations?) in the revised version with presented the statistical input of multiple localisation. Without allowing multiple localisations (we have done it by removing all metabolites that show multiple localisation), intra-compartment inhibition would be even more significant as in the real network (Supplementary Figure S7a).

Figure 5a**Figure S7a**
Response Figure 5: Computational simulation of the effect of compartmentalisation on the enzyme inhibition. (Figure 5a) Computational simulation of the effect of a hypothetical random compartmentalisation on the enzyme inhibition network, based on randomly placing inhibitors and enzymes in random compartments which includes multiple localization. (Figure S7a) Computational simulation of the effect of compartmentalisation on the enzyme inhibition, based on randomly placing inhibitors and enzymes in any single random compartment. The red line represents bs spline fitting.

(3) The authors also find a significant correlation between the number of inhibitory reactions and the number in which a given metabolite participates. However, I suspect that this correlation is the result of the above mentioned chemical similarity between compounds in adjacent reactions. If the authors want to make the point, they should try to detach both effects.

The reviewer is right that the high correlation between the number of metabolic reaction and the number of inhibitory interactions is due to high structural similarity of inhibitor and compounds from its adjacent metabolic reactions. This, in fact, is one of the key results of the paper which highlights that the compounds from adjacent metabolic reactions of an inhibitor are its potential competitors at the enzyme level – due to structural similarity. Therefore inhibition is, to a substantial extent, a topological structural property of the metabolic system. Please see the reply to reviewer #1, indeed essential metabolites are more structurally similar to one another, as non-essential metabolites, and this is one of the reasons why they are most likely an inhibitor. This is, as secondary metabolites are more peripheral to the metabolic system and have undergone more enzymatic steps. The text has been adjusted accordingly.

The compartmentalisation analysis as mentioned above disentangles the scenarios of metabolite structural similarity and the pressure for being an inhibitor. We would like to Refer to the reply to Reviewer #1.

(4) Finally, the authors discuss a potential lack of coherence between reported enzymatic constants (K_m and K_i) and the real inhibitory effect of metabolites *in vivo*, and they speculate that strong inhibitory constants might become irrelevant if the enzymes and metabolites are in different compartments and vice versa. I agree with their observation that quantitative data is highly heterogeneous and incomplete, however, it would be interesting to see if there is any correlation between K_m/K_i reported constants within or between compartments and well documented cases of *in vivo* inhibition, and see if any sort of “biologically relevant” inhibition threshold could be defined.

This would be an excellent suggestion and we would love to define such ‘cut-off’, although this would need to be a metabolite-specific value; however, the problem is that the same reaction kinetic constants (=same enzyme, same metabolic inhibitor) reported by different laboratories vary in many case by orders of magnitude (Suppl Figure S8). For the absolute quantitative values, the experimental difference from the lab to the lab are hence much stronger as the biological effect that can be expected. Hence, for this technical reasons, we can not calculate this ‘threshold’, as the Brenda quantitative data is by far too noisy for a global-scale, quantitative analysis cross-species..

Reviewer #3

In this study, Alam et al. use the vast information accumulated in the last century on enzyme inhibition to draw a semi-global network of metabolites and the enzymes they inhibit. The authors analyze this inhibition network and uncover multiple interesting phenomena that directly relates to our understanding of the function and evolution of metabolism. The work described in this study is innovative, provocative and of interest for very broad range of readers. However, it does suffer from several problems, which I list below. Do these problems prohibit the publication of the study? Honestly, my mind is split here and I will let the editor decide.

We thank the reviewer for a positive general statement and considering our study ‘innovative, provocative and of interest for a very broad range of readers’. We will list below how we have dealt with all the statistical challenges in making sense out of an

indeed very heterogeneous enzyme inhibitor data, derived from a century of biochemical research, and will address in detail how we dealt with all instances that the reviewer brought up.

1. Multiple measurements biases might affect, and ultimately be responsible for, many of the trends noticed by the authors. While the authors are aware of it, and refer to this issue in multiple places, the explanations they provide for justify the results are not fully convincing.

Before going into the detailed point-to-point reply, we would like to take the opportunity for a general reply to this comment. As the Reviewer notes, the authors of this manuscript are aware of the limits of the inhibitor datasets as determined by enzymologists in thousands of individual studies. The Reviewer acknowledges that we have listed all restrictions in detail. Indeed, both lead author and the second senior author have been made aware of these problems throughout their careers, as both did not start their career in computational systems biology but instead in structural biology (BFL) or in gene-centric yeast molecular biology (MR)).

The problem the reviewer addresses is, hence, not so much about the question of awareness of the authors, but it is whether the enzymological data generated by three generations of biochemists *in vitro* is, in principle, good enough - or too much biased - to draw any general conclusions. If the bias is that enzymological data are too big and noisy to be solved, then it would, in essence, mean to ignore the century of biochemical work and not to feed any of this existing enzymological knowledge into system scale studies that enable general conclusions. However, *if* the bias can be dealt with some powerful statistical methods, then it might lead to the general principles of enzyme function in the cell which are scientific, and at present, unanswerable otherwise.

The authors of this study think that the work of enzymologists is highly valuable and should not be ignored for work on the network scale, despite certainly not all data points are a perfect match to one another and there are definite limits about *in vitro* enzymology. We hence have chosen to work with what is possible at present. In order to work around the existing limits, we applied several statistical methods and random simulations, and address the bias, and where it is not possible to alleviate bias, we try to identify and quantify the existing limits. Finally please note, the points risen in Reviewer #3 comment #1 are **not limits of our study** in particular, but of enzymology in general. They will hence affect other work as much as they affect ours. This includes also smaller-scale studies, that may take just a few datapoints from Brenda and not all as we do. Indeed, we think this is in fact a strength of our study: By working with all enzymatic

data, we feel in a qualified position to comment about both the strengths, but also the weakness of *in vitro* enzymology conducted enzyme by enzyme, and we have done so extensively in the discussion and in the results sections

Below, we explain in detail how we addressed the specific problems the reviewer lists (all of these were known to us and are addressed in the manuscript).

Detailed reply to the sub points of Reviewer #3 comment 1, about the statistical and coverage bias that exists within enzymological data, and how we identify and dealt with the bias

1.1 “Each enzyme class is comparably well covered by inhibitor information, ruling out that enzymological data does bias against a certain type of enzyme” – the fact that all enzyme classes are represented does not mean that there is no bias towards specific types of enzymes – some might still be measured many times while others only few times.

We have addressed and discussed this potential bias in our manuscript. The ~200,000 manuscripts collected in Brenda cover the enzymatic reactions of the metabolic reconstruction to the extent that all enzyme classes are similarly well covered with inhibitor information.

Response Figure 6: Percent coverage of enzyme classes within the inhibition network with respect to the total enzymes of the human metabolic network reconstruction recon2 for each enzyme class.

1.2. “By far the largest total number of inhibitions are mediated by the HMDB superclass Nucleosides, Nucleotides, and Analogues ...” – this could be the results of measurement bias, as ATP and analogs were probably tested much more than any other type of compounds. “Despite some metabolites, like for instance ATP, have certainly been more studied than other metabolites, they act as typical inhibitors on a subset of enzyme classes...” – again, can’t it be that for specific enzyme classes, like ligases, which utilize nucleotides as substrates, “Nucleosides, Nucleotides, and Analogues” were tested as inhibitors more frequently than for other enzyme classes?

1.3. “...and that clear patterns emerge, despite the potential coverage bias, caused by the fact that the inhibition network was assembled from small scale enzymological data.” – I still don’t understand how the coverage bias is factored out here.

We have addressed this problem in the manuscript. It is well possible that ATP, ADP and AMP may have been over proportionally tested. But the enrichment analysis conducted corrects for the sample size. Further, the statement that they represent the most frequent inhibitor class holds true, which is shown by at least two other results that are insensitive to a potential testing bias as they base on the metabolic network topology and not on the BRENDA dataset. First, ATP, ADP, and AMP are not only among the most common inhibitors, but also among the most connected metabolites in the metabolic network (they play a role in a higher number of biochemical enzymatic reactions) (Figure 4f). Therefore, metabolites that are structurally similar to one of these (i.e., the other ‘Nucleosides, Nucleotides, and Analogues’ Figure 4a, b) have a much higher chance to be a competitive inhibitor. So there is a biological reason why this is the most frequently enzyme-bound metabolite class in the network, which actually implies that they should be the most frequent inhibitor class as well (we show that metabolites within one chemical class are more structurally similar to one another, a finding which is intuitive). Aside, they are highly concentrated in the cell, which amplifies the importance of this finding. Second, the specificity of the result is reflected that *Nucleosides, Nucleotides, and Analogues* are specifically enriched to inhibit i.e. a large enzyme class, *Transferases* (Figure 1f). A testing bias, as questioned by the Reviewer, could not explain a specific enrichment of the Nucleotides to inhibit this class of enzymes - it would affect all enzyme mechanisms more or less equally and hence not result in a highly significant enrichment (given the richness of the data resource). Please note that the used statistical tests correct for sample size. The obtained significance P-value for the enrichment of this HMDB class to inhibit transferases is $4.8e-34$. Further, we detect highly specific enrichments for ALL HMDB metabolite superclasses, and enzyme classes (Figure 1). Its hence the chemical nature of the metabolite which determines the type of enzyme it inhibits. We have intensively re-worked the respective paragraph to make this clearer.

1.4. “the closer the metabolite is in the metabolic network to the substrate of the enzyme (within the same metabolic pathway), the more likely it has been discovered as inhibitor for the particular enzyme by a study listed in BRENDA” – this could be a classic measurement bias; enzymologist characterizing an enzyme were biased to test other metabolites in the pathway for inhibition.

We can fully rule out this result to be solely the cause of a testing bias: These inhibitors are not only close neighbors in the metabolic system, but – according to all five applied structural similarity algorithms – they possess a significantly higher structural similarity, which is a key condition for being a competitive inhibitor. Further, we show that structurally similar metabolites have a closer distance in the network, and metabolites within the same pathway have a significantly higher structural similarity to one another. This analysis was done by using the topology of the metabolic network reconstruction, and is hence not influenced by the enzyme inhibitor dataset coverage. Hence, a coverage bias does not explain a significant higher structural similarity in dependency of network distance.

1.5. “we found that the central metabolic pathways glycolysis, gluconeogenesis and nucleotide interconversions, were the most inhibited and inhibiting metabolic pathways” – could be another measurement bias. The authors try to explain, in the following lines, why this is no so, but I failed to understand the explanation. [for example, what is an “inhibitory reaction”? Could be either an inhibitory compound or an inhibited reaction...]

“the large amount of enzyme data covered the metabolic network reconstruction to the extent that we find no selective bias against a certain enzyme class or reaction mechanism, even though some have been clearly more studied than others” – I again fail to understand this logic.

We apologize if this paragraph was misleading. It’s now improved; inhibitory ‘reaction’ should have meant inhibitory interactions (i.e. enzyme inhibited by a metabolite). We had discussed at length that this particular result – glycolytic enzymes to be among the most frequently inhibited

metabolites – could indeed be caused by a testing bias. In the manuscript, we continued with a downstream analysis in which we expanded the analysis to the entire metabolic network. While we cannot exclude the testing bias for glycolysis and the PPP, we can exclude it for the analysis that includes the entire metabolic network. Across the metabolic landscape we obtained a similar picture as implied by glycolysis (that the central metabolites are more likely to inhibit an enzymatic reaction). This result is not only statistically highly significant, but the authors find it also intuitive, for instance, as secondary and distal metabolites can be easier changed over the evolutionary timeline. As this genome-scale analysis leads to a similar conclusion considering the most central metabolites, it is very likely that also this result for glycolytic intermediates holds true. We have carefully re-worded that paragraph to avoid confusion or overstatements.

1.6. The fact that there exist a lot of data does not mean that there are also significant measurement biases.

We fully agree. But the error typically scales with sample size. For this reason, we have – where appropriate – corrected all our statistical results for a false discovery rate (FDR) and work typically with enrichment statistics that takes sample size into account..

1.7. “as a rough proxy of metabolite’s market properties, we plotted the price of the chemicals in the 2016 Sigma-Aldrich catalogue against the number of associated inhibitory reactions, and found no correlation” –

We disagree with the reviewer that this analysis to be obsolete. Many laboratories around the world have a tight budget. We hence had to test whether the purchase price of chemicals biases their coverage within the inhibition network. We considered the Sigma-Aldrich catalogue to be a sensible compromise as it provides a large enough dataset to test for a commercial bias. There was, however, no correlation found, so a metabolite purchase cost can most likely to be ruled out as a potential confounding factor within the network. As the reviewer finds this less important, we have however now moved the analysis to a less visible place in text and supplement.

1.8. “The nature of this network shows that despite the coverage bias that exists within the existing enzymological data, a clear preference between metabolic inhibitor and the affected enzyme classes is pictured.” – again I fail to see the argument here – see above.

We would like to refer to the above-mentioned general comments to address this concern. While we agree with the Reviewer that data generated by enzymologist has its limits, we think that careful statistical analysis, as conducted in our study, and massive size of the dataset combining ~200,000 studies (!) can make sense out of this data despite there is bias in the literature. We come to several new and important general conclusions that would not be derived by looking only at single reaction, enzyme, or inhibitor at a time.

2. Why did the authors ignore the list of activator compounds? These are also given in BRENDA and are equal in importance. In a sense, inhibitors and activators are the same thing with a reverse sign, and to get a full picture of the inhibition/activation network, they should be included. Of course, considering compounds that bind to the same active site as the substrate, only inhibitors make sense, but for allosteric effects, inhibitors and activators serve the same basic role. The authors say: “Due to the presence of inhibitory metabolites for virtually each enzymatic reaction, the activity of many metabolic enzymes in vivo would be lower than expected from in vitro kinetics” –

however, this is not true when activators are considered. Indeed, the measured k_{cat} of many enzymes (e.g., *E. coli*'s SerA) is far too low to explain their *in vivo* activity and hence there must be some activators that significantly enhance their activity *in vivo*.

We agree with the Reviewer about the importance of activators. Our current manuscript, addressing enzyme inhibitors on the genomic scale, contains however already more than 50 figure panels, each of these with its underlying statistics, an extensive discussion, and extensive supplementary data materials. The Brenda-stored data about enzyme activators has different constraints, different biases, different pitfalls, and needs different statistical strategies to be addressed in a systematic fashion. In particular, as the quantitative information about activators is much less comprehensive as the one about the inhibitors, and hence suffers from a much stronger coverage bias as the inhibitor data used herein (please see the other comments, the Reviewer himself addresses in most of his comments the coverage bias to represents to be the considerable difficulty that had to be overcome in analysing the enzymological data systematically). To include enzyme activators comprehensively hence requires at least a doubling of the current figures, new statistics to be introduced to deal with the additional constraints, and will lead to at least a doubling of introduction, results and discussion section, as several new constraints need to be introduced and explained. We believe that doubling our already very comprehensive manuscript would impact significantly on its focus, make it much more difficult to read and comprehend, and hence, in essence, considerably reduce the impact of the paper. We hence consider a genome-scale network of enzyme activators outside of the focus of this study. However, we appreciate the comment of the reviewer as a very valuable suggestion for future work leading to another manuscript. We included a sentence to the discussion, that alike for enzyme inhibitors, the genome-scale importance of enzyme activators is not understood, and warrants a systematic investigation.

Minor concerns

1. From previous experience I had with the BRENDA database I found that ~20% of the kinetic data reports are erroneous, i.e., do not fit the data reported in the corresponding papers, due to mistakes in copying on unit mismatch. While I think that the inhibition data is more accurate it is still worthwhile reporting the errors founds and how a general analysis could deal with such errors.

While we ourselves can not reproduce an error rate as high as 20% in the Brenda database, we agree that there are certainly 'copy-paste' problem from the primary

literature, leading to errors in all enzymological data collections including the Brenda database. 'Unit mismatch' as the ones mentioned by the reviewer do not affect our manuscript however, as we work in a qualitative and not on a quantitative network for the reasons explained in the comments above. Typos in the names of enzymes and metabolites have been addressed in the extensive curation strategy. Most of these errors are eliminated by computationally matching EC numbers and metabolites to PubChem, KEGG and HMDB identifiers and mapping them to the human metabolic reconstruction, recon2. All metabolite names and enzymatic reactions that have not a unique match with the metabolic network reconstruction were discarded. Further, we manually addressed all other entries, which eliminated further association errors. We have improved the description of the curation paragraph, to explain the situation.

2. The authors claim that enzyme inhibition is typically not specific to a given organism and hence it is justified to integrate data from other organisms to create the human inhibitory network. I'm not convinced that this is indeed the case. For example, multiple enzymes that are represented by the same EC number, actually corresponds to different catalytic mechanisms, or 'Classes', such that we (human) harbor an enzyme from one class and another organism harbors an enzyme from a different class – both are completely different in mechanism and inhibition.

We apologize that the reviewer was misled by our explanation of the aim of our study. We did not attempt to create a species specific, 'human only' network. This is not possible at present, for the simple reason that also the 'human' metabolic reconstruction does build on enzymological information recorded across species (Thiele et al. 2013), and that -by far- there is not species-specific enzymological data out there for doing so. Recon2 is called 'human' as it includes the metabolic reactions that are believed to be part of the human metabolic network; and in cases where there are multiple topological options, it chooses the 'human' version. We have taken the same strategy in our manuscript, as the most sensible compromise. The basic assumption to justify this strategy is that the principles that underlie enzyme inhibition are the same across species (as they build on the same chemistry). Means, even if our network contains a 'false positive' inhibition reaction in the sense that this inhibition is in fact happening in another species and not in a human cell, does not have a negative impact on our general conclusions, as long as human and 'non-human' enzyme inhibition do base on the same general principles. We have re-written the text accordingly, making it clear that we work on a 'cross species' inhibition network to uncover basic principles of enzyme inhibition. The reason we use the human reconstruction is that it is one of the largest and best curated models at present, and as we can access the chemical

categorisation of HMDB to group inhibitors, please the reply to comment #3 and comment #5 as well.

Please also note that Reviewer #1 highlights exactly this strategy as the reasonable choice.

3. The inhibitors were classified to different groups. However, the presence of phosphate was not taken into account. For example, I think that differentiating between “Carbohydrates” and “Phosphorylated-Carbohydrates” might be highly interesting as it is well-known that protein binding to phosphate groups has a special significance.

As the reviewer might acknowledge, categorizing metabolites in chemical group is not a trivial task, but one that has been elegantly solved by HMDB by using dozens of chemical rules in how to define a group of metabolites, of which phosphorylation is *ONE* of many criteria. For this reason, characterization has become a *de-facto* community standard. We have used the canonical metabolite grouping of HMDB first as it is an excellent compromise to do chemical categorisation, and second as it enables other researchers to download and work with our network, as it retains compatible with other studies. The re-usability of our dataset by other researchers and the fact that it adheres with community standards is a key impact generating factor of our manuscript. Hence, we think the best option is to not touch the chemical grouping for the purpose of our study, and please note, that chemical composition has been part of the HMDB rules indeed.

In response to the reviewer, and fully acknowledging the importance of phosphorylation, we have calculated the number of phosphorylated inhibitor compounds for each group, and discussed that in the manuscript (See new Supplementary Table S2).

In the result section: 29% (197/682) of inhibitors in the network are phosphorylated. This includes 83.3% of the metabolites in the category Nucleosides, Nucleotides and Analogues, 36.5% of Carbohydrates and Carbohydrate Conjugates, 26.4% of Lipids and 3.5% in the group of Amino Acids, Peptides and Analogues category (Supplementary Table S2).

4. “33% of noncompetitive and 31% uncompetitive did however possess significant structural similarity to the enzyme’s substrate (Fig 2b)” – I find this statement a bit problematic as the distributions in Figure 2b seems rather spread over all the similarity

range. Hence, it seems that the structural similarity plays no actual role in allosteric inhibitors, where some, just by mere chance, are similar to the substrate. I do not dispute the results but simply the way they are presented: instead of saying that 2/3 are not similar and a 1/3 is similar, I would say that it seems that similarity plays little role in allosteric inhibitors.

We apologize that our figure may have been misleading by using a boxplot, which does indeed not illustrate the distribution density. The numbers (33% and 31%, respectively) are derived from a calculation of the metabolite structural similarity by the multiple algorithms as described. We have checked the calculations and found them correct. The boxplots in Figure 2b and Supplementary Figure S1 have now been replaced by a violin plot that shows the densities, and we hope this illustration makes the result much clearer. Furthermore, we have revised the text to make misinterpretation less likely.

Response Figure 7: (Figure 2b) Pairwise compound similarity (fpSim) between inhibitor and substrate reveals significant similarity between competitive inhibitors and at least one metabolic substrates (0=non-similar, to 1=most similar). The median similarity for allosteric inhibitors is not significantly different to that of a random inhibitor, however, the spread is much higher, with a subset of allosteric inhibitors being highly similar to the enzyme's metabolic substrates. **Figure S1.** Pairwise compound similarity between inhibitor and substrate for competitive, noncompetitive and uncompetitive type of inhibition, compared to similarity between pairs of random compounds, using the following computational similarity coefficients: (a) Drug molecule similarity derived by molecular fingerprints (calcDrugFPSim (D.-S. Cao et al. 2015)), (b) Pairwise compound comparisons with atom pairs (cmp.similarity (Y. Cao et al. 2008)), (c) Tanimoto Coefficient for Maximum common substructure (fmcsR (Wang et al. 2013)) (d) Overlap Coefficient Maximum common substructure (Wang et al. 2013).

5. "...and use the human metabolic network (recon2), one of the most comprehensive metabolic network reconstructions, as a basis of our analysis." – here I wonder why the

authors did not try to further use *E. coli* or yeast to strengthen their findings – there is vast data on the enzymes of both of these microbes.

The reviewer is right that our analysis could have been done on all metabolic network reconstructions, including the one of *E. coli*, as we take only the chemical topology from the network. However, please see above, none of them is generated from species specific data, so in essence this part makes little difference, as the topology is highly redundant. Also the models have build on one another; the first yeast model was made from the *E. coli* model, the early human model did build on the yeast model.

We have chosen the human model as it is a) one of the most comprehensive community reconstruction to date, and b) as we think that our study provides the best impact among the readership of *Nature Communications* if centering around human metabolism. Further, this model has been generated with a particular emphasis on the subcellular compartmentalization in organelles. This is a key aspect of our paper, and could not have been done on the *E. coli* model which does not include such compartmentalisation, nor metabolic cooperativity between bacterial cells/species into account either.

Nonetheless, we did also use *E. coli* model to take its non-human part, and by combining it with the human model to examine how the bacterial metabolites would inhibit human enzymes and *vice versa*. We do show the principal outcome of the network joining in the discussion. Further we have mapped the inhibitors also on the yeast network, and although smaller as the human reconstruction, we obtain a similarly structured inhibition network. This is however expected/trivial, as the core metabolic network models of yeast and human possess a very similar topology.

Indeed, at the early stage of our work we did, as the Reviewer, consider the possibility of a strictly 'species specific' network. The reviewer is right that *E. coli* and yeast would be the appropriate choices for such a network. However, even though for *E. coli* and yeast there is a decent amount of inhibitor information available, the species specific dataset is negligibly small if compared to the 200,000 studies that we compiled into our 'cross species' inhibition network: While we agree with the Reviewer to 100% that it would have been nice work on data recorded on one species only, for our study there is simply not enough data out there (To illustrate the problem: until recently (2013) there was not even enough data to model just glycolysis, the best studied metabolic pathway from yeast only data (Cornish-Bowden et al. 2013)) For our study to address general principles behind enzyme inhibition, we need the massive cross-species network coverage to alleviate the measurement biases discussed above, and to have enough

data points for the machine-learning based structural algorithms, enabling us to reveal the general principles behind enzyme inhibition. We agree that other studies, i.e, address metabolic regulation which is much more species specific, quantitative, and perhaps less sensitive to obtain a comprehensive network coverage, might be better conducted with smaller, species specific networks. But the enzymological data coverage is highly limiting to do any 'one species only' work on a network scale.

6. "It follows that a majority of enzyme inhibition by cell's metabolites did not evolve to regulate metabolism. Instead, enzyme inhibition emerges for structural reasons, which implies that many inhibitory interactions have negative impact on the functionality of metabolic network." – this is an interesting assumption to which I tend to agree. Still I do not think that the authors "proved" it and hence I would state it as a mere assumption.

We apologize if this was considered an overstatement, and have re-written this paragraph.

7. "one has to take into consideration that the metabolites which are part of the modern metabolic pathways are the ones that prevailed during evolution, while strong inhibitors like the aforementioned 4-phospho-erythronate and 2-phospho-L-lactate had to be removed from the system in order for cells to survive" – I would be very careful here as there are highly reactive/toxic/inhibitory compounds which are used as metabolites by some organisms and at some conditions; for example, formaldehyde, methyglyoxal, and hydroxyserine.

We agree with the reviewer about the principle he mentions; however, we have not found any literature which would show any species using 4-phospho-erythronate or 2-phospho-L-lactate as part of their normal metabolism, yet of course, these could exist and be discovered in the future. As compromise, we have replaced the word 'modern metabolic network' with 'human metabolic network reconstruction', in this way the statement correctly quotes the literature that discovered these metabolic as inhibitors for human and yeast metabolism, and the constraint they impose on metabolism of that particular species.

Reviewer #4:

The manuscript by Alam et al. attempts to provide global analysis of the metabolome and its constrains. Understanding cellular networks and their regulations,

interconnectivity and constrains is key to physiology and biology. The authors focus on non-catalytic metabolite enzyme interactions, which may play a role in major regulatory functions. This is very interesting manuscript but quite uneven.

(1) The authors merged enzyme knowledge accumulated over the past 100 years with genome-based metabolite enzyme-inhibitor networks. To validate their hypothesis, they use published and new structural studies of allosterically inhibited L-lactate dehydrogenase. Unfortunately they were unable to compare K_i/K_m for these enzymes and metabolites. *E. coli*. *E. coli*.

Unfortunately we were unable to compare the ratio K_i/K_m for all the enzymes and metabolites due to the high variation in these kinetic constants reported for the same enzyme by different laboratories (Comments above and Suppl Figure S8).

However, what we were able to compare was the K_i/K_m for L-LDH, where we included the comparison of the K_i value for oxaloacetate and the non common intermediate malonate versus the K_m for pyruvate. This shows that even though both intermediates are not strong inhibitors, the oxaloacetate, that is a common intermediate in the cell, has more inhibitory capacity compared to malonate, for which the mammalian L-LDH did not evolve to be inhibited.

(2) Here I comment on the quality of the crystal structures reported in the paper. The crystal structure of rabbit muscle L-LDH was determined as a complex with oxaloacetate and NADH cofactor and with malonate. These two structures are high resolution and should be very similar but showed some different stats. It is quite difficult to judge just based on Table S4 because I do not have access to the data and PDB report. I do not question the validity of the structures (unfortunately no electron density is available), but there are questions on how these structures have been done. For example, the higher resolution structure has poorer rms on bond length and angles. Why?

The malonate complex was refined with *refmac*, and we have run the refinement through another cycle of *PHENIX* refinement and revised the tables. The rms bonds and angles are 0.010 angstroms and 1.398 degrees. The maps are very clear and according to the the state of the art, and reveal many unbiased features that were not part of the original model, such as positions of ligands, waters, multiple conformations of side chains. Please note that the structures PDB deposits will be released with publishing of the article and we hope this additional features, that are not relevant for our study, will help other researchers that are interested in L-LDH.

(3) There are some inconsistent things with the data and refinement statistics. It

appears that the lower resolution structure is "better" than the higher resolution structure. The higher resolution structure has significantly less observations (less than half) than the lower resolution structure. Why is that? The coverage is poorer for higher resolution structure as well as I/σ . At the same time more unique reflections are used for refinement, which is quite strange.

The lower resolution data was collected with high redundancy, while the higher resolution structure had longer exposure times and less redundancy due to radiation damage. We have now added a comment to the materials and methods with this caveat.

(4) Even stranger is the fact that these structures seem to be quite an important part of the observations but they are not discussed in "Discussion" section. Moreover, there are a number of crystal structures of human and rabbit LDHs but they are not mentioned at all.

The structures are now discussed in the revised discussion.

(5) In general this is very interesting manuscript but it is quite sloppy and there are a number of minor issues with arguments, text and English. One example is: "studied by X-ray diffraction" rather than "studied by X-ray crystallography"

We apologize that language and grammar inconsistencies. We have intensively re-worked the language of the manuscript, in particular with the help of the native speaking co-authors of the study. We hope the edits proof is satisfactory.

References

- Campbell, Kate, Jakob Vowinckel, Michael Mülleder, Silke Malmshheimer, Nicola Lawrence, Enrica Calvani, Leonor Miller-Fleming, et al. 2015. "Self-Establishing Communities Enable Cooperative Metabolite Exchange in a Eukaryote." *eLife* 4 (October). doi:10.7554/eLife.09943.
- Cao, Dong-Sheng, Nan Xiao, Qing-Song Xu, and Alex F. Chen. 2015. "Rcpi: R/Bioconductor Package to Generate Various Descriptors of Proteins, Compounds and Their Interactions." *Bioinformatics* 31 (2): 279–81.
- Cao, Yiqun, Anna Charisi, Li-Chang Cheng, Tao Jiang, and Thomas Girke. 2008. "ChemmineR: A Compound Mining Framework for R." *Bioinformatics* 24 (15): 1733–34.
- Chandra, Fiona A., Gentian Buzi, and John C. Doyle. 2011. "Glycolytic Oscillations and

- Limits on Robust Efficiency." *Science* 333 (6039): 187–92.
- Chen, Anna H., and Pamela A. Silver. 2012. "Designing Biological Compartmentalization." *Trends in Cell Biology* 22 (12): 662–70.
- Cornish-Bowden, Athel, Christian P. Whitman, Kieran Smallbone, Hanan L. Messiha, Kathleen M. Carroll, Catherine L. Winder, Naglis Malys, et al. 2013. "A Model of Yeast Glycolysis Based on a Consistent Kinetic Characterisation of All Its Enzymes." *FEBS Letters*.
<http://www.sciencedirect.com/science/article/pii/S0014579313005012>.
- Giessen, Tobias W., and Pamela A. Silver. 2016. "Encapsulation as a Strategy for the Design of Biological Compartmentalization." *Journal of Molecular Biology* 428 (5 Pt B): 916–27.
- Liu, Yuchen, Laura L. Beer, and William B. Whitman. 2012. "Methanogens: A Window into Ancient Sulfur Metabolism." *Trends in Microbiology* 20 (5): 251–58.
- Mee, Michael T., James J. Collins, George M. Church, and Harris H. Wang. 2014. "Syntrophic Exchange in Synthetic Microbial Communities." *Proceedings of the National Academy of Sciences of the United States of America* 111 (20): E2149–56.
- Thiele, Ines, Neil Swainston, Ronan M. T. Fleming, Andreas Hoppe, Swagatika Sahoo, Maïke K. Aurich, Hulda Haraldsdottir, et al. 2013. "A Community-Driven Global Reconstruction of Human Metabolism." *Nature Biotechnology* 31 (5): 419–25.
- Tu, Benjamin P., Andrzej Kudlicki, Maga Rowicka, and Steven L. McKnight. 2005. "Logic of the Yeast Metabolic Cycle: Temporal Compartmentalization of Cellular Processes." *Science* 310 (5751): 1152–58.
- Wang, Yan, Tyler W. H. Backman, Kevin Horan, and Thomas Girke. 2013. "fmcsR: Mismatch Tolerant Maximum Common Substructure Searching in R." *Bioinformatics* 29 (21): 2792–94.
- Wintermute, Edwin H., and Pamela A. Silver. 2010. "Emergent Cooperation in Microbial Metabolism." *Molecular Systems Biology* 6 (September): 407.
- Xu, Peng, Sridhar Ranganathan, Zachary L. Fowler, Costas D. Maranas, and Mattheos A. G. Koffas. 2011. "Genome-Scale Metabolic Network Modeling Results in Minimal Interventions That Cooperatively Force Carbon Flux towards Malonyl-CoA." *Metabolic Engineering* 13 (5): 578–87.
- Zelezniak, Aleksej, Sergej Andrejev, Olga Ponomarova, Daniel R. Mende, Peer Bork, and Kiran Raosaheb Patil. 2015. "Metabolic Dependencies Drive Species Co-Occurrence in Diverse Microbial Communities." *Proceedings of the National Academy of Sciences* 112 (20): 201421834.

REVIEWERS' COMMENTS:

Reviewer #1 (Remarks to the Author):

I found the revised version of the manuscript substantially improved and the authors made a great job to address my previous concerns.

I have only two minor comments left:

First, some P-values are still reported without the corresponding statistical test in the Results section (e.g. see page 14, 15). Even if the test is mentioned elsewhere, it should be also mentioned right next to the P-value to allow the reader to easily understand the results.

Second, the manuscript still contains some typos:

- Typo at the bottom of page 4: "adress" in "Corrected by statistical methods to adress the bias that exists in enzymological data"
- page 5: " As a consequences"
- page 6: " Aa minority of enzymatic reactions"; "and the range than spans up"; " that are participating also als metabolites"
- page 7: "1990ies"
- page 14: "We metabolites clustered"
- page 21: "...by the evolution of a clearance enzyme cells"

Reviewer #2 (Remarks to the Author):

The authors have done a good job addressing my (and the other Reviewers') comments and, although not all the results go in the direction that the original m/s suggested, I believe that the work is more robust now. And I'm happy to recommend its publication in Nat Commun.

Reviewer #3 (Remarks to the Author):

Dear editor and authors,

Reading the authors' response and the corrections made to the manuscript, I am fully satisfied and support the publication of the study in Nature Communications.

Congratulations for the nice work!

Reviewer #4 (Remarks to the Author):

I am satisfied with authors response to my concerns.

Minor

There is error in the sentence "Instead, heir binding and effect on the enzyme .."

Reply to the Reviewers comments

We thank all four reviewers to approve our revision. We have corrected the additional typos spotted by two of the Reviewers, and have included the additional information about the t-tests used in the Figure legends.